# Pseudo-Riemannian Graph Convolutional Networks

**Bo Xiong**[*]
University of Stuttgart
Stuttgart, Germany

**Shichao Zhu**[*]
IIE, Chinese Academy of Sciences
Beijing, China

**Nico Potyka**
Imperial College London
London, United Kingdom

**Shirui Pan**
Griffith University
Queensland, Australia

**Chuan Zhou**
AMSS, Chinese Academy of Sciences
Beijing, China

**Steffen Staab**
University of Stuttgart
University of Southampton
Stuttgart, Germany

## Abstract

Graph convolutional networks (GCNs) are powerful frameworks for learning embeddings of graph-structured data. GCNs are traditionally studied through the lens of Euclidean geometry. Recent works find that non-Euclidean Riemannian manifolds provide specific inductive biases for embedding hierarchical or spherical data. However, they cannot align well with data of mixed graph topologies. We consider a larger class of pseudo-Riemannian manifolds that generalize hyperboloid and sphere. We develop new geodesic tools that allow for extending neural network operations into geodesically disconnected pseudo-Riemannian manifolds. As a consequence, we derive a pseudo-Riemannian GCN that models data in pseudo-Riemannian manifolds of constant nonzero curvature in the context of graph neural networks. Our method provides a geometric inductive bias that is sufficiently flexible to model mixed heterogeneous topologies like hierarchical graphs with cycles. We demonstrate the representational capabilities of this method by applying it to the tasks of graph reconstruction, node classification and link prediction on a series of standard graphs with mixed topologies. Empirical results demonstrate that our method outperforms Riemannian counterparts when embedding graphs of complex topologies.

## 1 Introduction

Learning from graph-structured data is a pivotal task in machine learning, for which graph convolutional networks (GCNs) [1, 2, 3, 4] have emerged as powerful graph representation learning techniques. GCNs exploit both features and structural properties in graphs, which makes them well-suited for a wide range of applications. For this purpose, graphs are usually embedded in Riemannian manifolds equipped with a positive definite metric. Euclidean geometry is a special case of Riemannian manifolds of constant zero curvature that can be understood intuitively and has well-defined operations. However, the representation power of Euclidean space is limited [5], especially when embedding complex graphs exhibiting hierarchical structures [6]. Non-Euclidean Riemannian manifolds of constant curvatures provide an alternative to accommodate specific graph topologies. For example, hyperbolic manifold of constant negative curvature has exponentially growing volume and is well suited to represent hierarchical structures such as tree-like graphs [7, 8, 9, 10, 11]. Similarly, spherical manifold of constant positive curvature is suitable for embedding spherical data in various fields [12, 13, 14] including graphs with cycles. Some recent works [15, 16, 17, 18, 19] have extended GCNs to such non-Euclidean manifolds and have shown substantial improvements.

---

[*]Equal contribution. Correspondence to bo.xiong@ipvs.uni-stuttgart.de

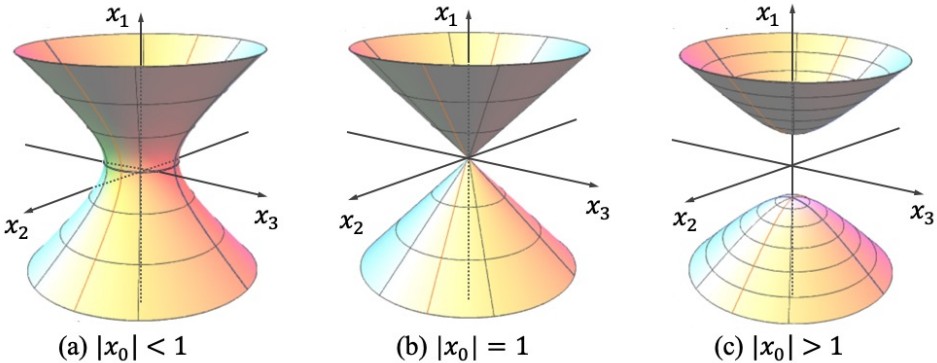

(a) $|x_0| < 1$        (b) $|x_0| = 1$        (c) $|x_0| > 1$

Figure 1: The different submanifolds of a four-dimensional pseudo-hyperboloid of curvature $-1$ with two time dimensions. By fixing one time dimension $x_0$, the induced submanifolds include (a) an one-sheet hyperboloid, (b) the double cone, and (c) a two-sheet hyperboloid.

The topologies in real-world graphs [6], however, usually exhibit highly heterogeneous topological structures, which are best represented by different geometrical curvatures. A globally homogeneous geometry lacks the flexibility for modeling complex graphs [20]. Instead of using a single manifold, product manifolds [20, 21] combining multiple Riemannian manifolds have shown advantages when embedding graphs of mixed topologies. However, the curvature distribution of product manifolds is the same at each point, which limits the capability of embedding topologically heterogeneous graphs. Furthermore, Riemannian manifolds are equipped with a positive definite metric disallowing for the faithful representation of the negative eigen-spectrum of input similarities [22].

Going beyond Riemannian manifolds, pseudo-Riemannian manifolds equipped with indefinite metrics constitute a larger class of geometries, pseudo-Riemannian manifolds of constant nonzero curvature do not only generalize the hyperbolic and spherical manifolds, but also contain their submanifolds (Cf. Fig. 1), thus providing inductive biases specific to these geometries. Pseudo-Riemannian geometry with constant zero curvature (i.e. Lorentzian spacetime) was applied to manifold learning for preserving local information of non-metric data [5] and embedding directed acyclic graph [23]. To model complex graphs containing both hierarchies and cycles, pseudo-Riemannian manifolds with constant nonzero curvature have recently been applied into graph embeddings using non-parametric learning [24, 25, 26], but the representation power of these works is not on par with the Riemannian counterparts yet, mostly because of the absence of geodesic tools to extend neural network operations into pseudo-Riemannian geometry.

In this paper, we take the first step to extend GCNs into pseudo-Riemannian manifolds foregoing the requirement to have a positive definite metric. Exploiting pseudo-Riemannian geometry for GCNs is non-trivial because of the *geodesical disconnectedness* in pseudo-Riemannian geometry. There exist broken points that cannot be smoothly connected by a geodesic, leaving necessary geodesic tools undefined. To deal with it, we develop novel geodesic tools that empower manipulating representations in geodesically disconnected pseudo-Riemannian manifolds. We make it by finding diffeomorphic manifolds that provide alternative geodesic operations that smoothly avoid broken cases. Subsequently, we generalize GCNs to learn representations of complex graphs in pseudo-Riemannian geometry by defining corresponding operations such as *linear transformation* and *tangential aggregation*. Different from previous works, the initial features of GCN could be fully defined in the Euclidean space. Thanks to the diffeomorphic operation that is bijective and differentiable, the standard gradient descent algorithm can be exploited to perform optimization.

To summarize, our main contributions are as follows: 1) We present neural network operations in pseudo-Riemannian manifolds with novel geodesic tools, to stimulate the applications of pseudo-Riemannian geometry in geometric deep learning. 2) We present a principled framework, pseudo-Riemannian GCN, which generalizes GCNs into pseudo-Riemannian manifolds with indefinite metrics, providing more flexible inductive biases to accommodate complex graphs with mixed topologies. 3) Extensive evaluations on three standard tasks demonstrate that our model outperforms baselines that operate in Riemannian manifolds. Source code is open available at https://github.com/xiongbo010/QGCN.

## 2 Preliminaries

### 2.1 Pseudo-Riemannian manifolds

A pseudo-Riemannian manifold [27] $(\mathcal{M}, g)$ is a smooth manifold $\mathcal{M}$ equipped with a nondegenerate and indefinite metric tensor $g$. Nondegeneracy means that for a given $\boldsymbol{\xi} \in \mathcal{T}_\mathbf{x}\mathcal{M}$, for any $\boldsymbol{\zeta} \in \mathcal{T}_\mathbf{x}\mathcal{M}$ we have $g_\mathbf{x}(\boldsymbol{\xi}, \boldsymbol{\zeta}) = 0$, then $\boldsymbol{\xi} = \mathbf{0}$. The metric tensor $g$ induces a scalar product on the *tangent* space $\mathcal{T}_\mathbf{x}\mathcal{M}$ for each point $\mathbf{x} \in \mathcal{M}$ such that $g_\mathbf{x} : \mathcal{T}_\mathbf{x}\mathcal{M} \times \mathcal{T}_\mathbf{x}\mathcal{M} \to \mathbb{R}$, where the *tangent* space $\mathcal{T}_\mathbf{x}\mathcal{M}$ can be seen as the first order local approximation of $\mathcal{M}$ around point $\mathbf{x}$. The elements of $\mathcal{T}_\mathbf{x}\mathcal{M}$ are called tangent vectors. *Indefinity* means that the metric tensor could be of arbitrary signs. A principal special case is the Riemannian geometry, where the metric tensor is positive definite (i.e. $\forall \boldsymbol{\xi} \in \mathcal{T}_\mathbf{x}\mathcal{M}, g_\mathbf{x}(\boldsymbol{\xi}, \boldsymbol{\xi}) > 0$ iff $\boldsymbol{\xi} \neq \mathbf{0}$).

**Pseudo-hyperboloid** By analogy with hyperboloid and sphere in Euclidean space. Pseudo-hyperboloids are defined as the submanifolds in the ambient pseudo-Euclidean space $\mathbb{R}^{s,t+1}$ with the dimensionality of $d = s + t + 1$ that uses the scalar product as $\forall \mathbf{x}, \mathbf{y} \in \mathbb{R}^{s,t+1}, \langle \mathbf{x}, \mathbf{y} \rangle_t = -\sum_{i=0}^{t} x_i y_i + \sum_{j=t+1}^{s+t} x_j y_j$. The scalar product induces a norm $\|\mathbf{x}\|_t^2 = \langle \mathbf{x}, \mathbf{x} \rangle_t$ that can be used to define a pseudo-hyperboloid $\mathcal{Q}_\beta^{s,t} = \left\{ \mathbf{x} = (x_0, x_1, \cdots, x_{s+t})^\top \in \mathbb{R}^{s,t+1} : \|\mathbf{x}\|_t^2 = \beta \right\}$, where $\beta$ is a nonzero real number parameter of curvature. $\mathcal{Q}_\beta^{s,t}$ is called *pseudo-sphere* when $\beta > 0$ and *pseudo-hyperboloid* when $\beta < 0$. Since $\mathcal{Q}_\beta^{s,t}$ is interchangeable with $\mathcal{Q}_{-\beta}^{t+1,s-1}$, we consider the pseudo-hyperbololid here. Following the terminology of special relativity, a point in $\mathcal{Q}_\beta^{s,t}$ can be interpreted as an event [5], where the first $t + 1$ dimensions are time dimensions and the last $s$ dimensions are space dimensions. Hyperbolic $\mathbb{H}$ and spherical $\mathbb{S}$ manifolds can be defined as the special cases of pseudo-hyperboloids by setting all time dimensions except one to be zero and setting all space dimensions to be zero, respectively, i.e. $\mathbb{H}_\beta = \mathcal{Q}_\beta^{s,1}, \mathbb{S}_{-\beta} = \mathcal{Q}_\beta^{0,t}$.

### 2.2 Geodesic tools of pseudo-hyperboloid

**Geodesic** A generalization of a *straight-line* in the Euclidean space to a manifold is called the *geodesic* [15, 28]. Formally, a geodesic $\gamma$ is defined as a constant speed curve $\gamma : \tau \mapsto \gamma(\tau) \in \mathcal{M}, \tau \in [0, 1]$ joining two points $\mathbf{x}, \mathbf{y} \in \mathcal{M}$ that minimizes the length, where the length of a curve is given by $L(\gamma) = \int_0^1 \sqrt{\left\| \frac{d}{dt} \gamma(\tau) \right\|_{\gamma(\tau)}} dt$. The geodesic holds that $\gamma^* = \arg\min_\gamma L(\gamma)$, such that $\gamma(0) = \boldsymbol{x}, \gamma(1) = \boldsymbol{y}$, and $\left\| \frac{d}{d\tau} \gamma(\tau) \right\|_{\gamma(\tau)} = 1$. By the means of the geodesic, the distance between $\mathbf{x}, \mathbf{y} \in \mathcal{Q}_\beta^{s,t}$ is defined as the arc length of geodesic $\gamma(\tau)$.

**Exponential and logarithmic maps** The connections between manifolds and *tangent* space are established by the differentiable exponential map and logarithmic map. The exponential map at $\mathbf{x}$ is defined as $\exp_\mathbf{x}(\boldsymbol{\xi}) = \gamma(1)$, which gives a way to project a vector $\boldsymbol{\xi} \in \mathcal{T}_\mathbf{x}\mathcal{M}$ to a point $\exp_\mathbf{x}(\boldsymbol{\xi}) \in \mathcal{M}$ on the manifold. The logarithmic map $\log_\mathbf{x} : \mathcal{M} \to \mathcal{T}_\mathbf{x}\mathcal{M}$ is defined as the inverse of the exponential map (i.e. $\log_\mathbf{x} = \exp_\mathbf{x}^{-1}$). Note that since $\mathcal{Q}_\beta^{s,t}$ is a geodesically complete manifold, the domain of the exponential map $\mathcal{D}_x$ is hence defined on the entire tangent space, i.e. $\mathcal{D}_x = \mathcal{T}_\mathbf{x}\mathcal{Q}_\beta^{s,t}$. However, as we will explain later, the logarithmic map $\log_\mathbf{x}(\mathbf{y})$ is only defined when there exists a a length-minimizing geodesic between $\mathbf{x}, \mathbf{y} \in \mathcal{Q}_\beta^{s,t}$. More details can be found in the Appendix C.

**Geodesical connectedness** A pseudo-Riemannian manifold $\mathcal{M}$ is *connected* iff any two points of $\mathcal{M}$ can be joined by a piecewise (broken) geodesic with each piece being a smooth geodesic. The manifold is *geodesically connected* (or *g-connected*) iff any two points can be smoothly connected by a geodesic, where the two points are called *g-connected*, otherwise called *g-disconnected*. Different from Riemannian manifolds in which the geodesical completeness implies the g-connectedness (Hopf–Rinow theorem [29]), pseudo-hyperboloid is a geodesically complete but not *g-connected* manifold where there exist points that cannot be smoothly connected by a geodesic [30]. Formally, in the pseudo-hyperboloid, two points $\mathbf{x}, \mathbf{y} \in \mathcal{Q}_\beta^{s,t}$ are *g-connected* iff $\langle \mathbf{x}, \mathbf{y} \rangle_t < |\beta|$. The set of *g-connected* points of $\mathbf{x} \in \mathcal{Q}_\beta^{s,t}$ is denoted as its *normal neighborhood* $\mathcal{U}_\mathbf{x} =$

$\left\{ \mathbf{y} \in \mathcal{Q}_\beta^{s,t} : \langle \mathbf{x}, \mathbf{y} \rangle_t < |\beta| \right\}$. For *g-disconnected* points $\mathbf{x}, \mathbf{y} \in \mathcal{Q}_\beta^{s,t}$, there does not exist a tangent vector $\boldsymbol{\xi}$ such that $\mathbf{y} = \exp_{\mathbf{x}}^\beta(\boldsymbol{\xi})$, which implies that its inverse $\log_{\mathbf{x}}^\beta(\cdot)$ is only defined in the *normal neighborhood* of $\mathbf{x}$. In a nutshell, the geodesic tools for the *g-disconnected* cases are not well-defined, making it impossible to define corresponding vector operations.

# 3 Pseudo-Riemannian GCNs

In this section, we first describe how to tackle the *g-disconnectedness* in pseudo-Riemannian manifolds. Then we present the pseudo-Riemannian GCNs based on the proposed geodesic tools.

## 3.1 Diffeomorphic geodesic tools

One standard way to tackle the *g-disconnectedness* in differential geometry is to introduce diffeomorphic manifolds in which the operations are well-defined. A diffeomorphic manifold can be derived from a diffeomorphism, defined as follows.

**Definition 1** (Diffeomorphism [27]). *Given two manifolds $\mathcal{M}$ and $\mathcal{M}'$, a smooth map $\psi : \mathcal{M} \to \mathcal{M}'$ is called a diffeomorphism if $\psi$ is bijective and its inverse $\psi^{-1}$ is smooth as well. If a diffeomorphism between $\mathcal{M}$ and $\mathcal{M}'$ exists, we call them diffeomorphic and write $\mathcal{M} \simeq \mathcal{M}'$.*

For pseudo-Riemannian manifolds, the following diffeomorphism [24] decomposes pseudo-hyperboloid into the product manifolds of an unit sphere and the Euclidean space.

**Theorem 1** (Theorem 4.1 in [24]). *For any point $\mathbf{x} \in \mathcal{Q}_\beta^{s,t}$, there exists a diffeomorphism $\psi : \mathcal{Q}_\beta^{s,t} \to \mathbb{S}_1^t \times \mathbb{R}^s$ that maps $\mathbf{x}$ into the product manifolds of an unit sphere and the Euclidean space.*

The diffeomorphism is given in the Appendix D.1. In light of this, we introduce a new diffeomorphism that maps $\mathbf{x}$ to the product manifolds of sphere with curvature $-1/\beta$ and the Euclidean space.

**Theorem 2.** *For any point $\mathbf{x} \in \mathcal{Q}_\beta^{s,t}$, there exists a diffeomorphism $\psi : \mathcal{Q}_\beta^{s,t} \to \mathbb{S}_{-\beta}^t \times \mathbb{R}^s$ that maps $\mathbf{x}$ into the product manifolds of a sphere and the Euclidean space (proof in the Appendix D.2).*

Compared with Theorem 1, this diffeomorphism preserves the curvatures in the diffeomorphic components, making it satisfy some geometric properties, e.g. the mapped point $\mathbf{x}_t \in \mathbb{S}_{-\beta}^t$ still lies on the surface of the pseudo-hyperboloid, making moving the tangent vectors from the pseudo-hyperboloid to the diffeomorphic manifold easy as we explained later. We call $\psi$ as the spherical projection.

**Exponential and logarithmic maps.** Considering that pseudo-hyperboloid $\mathcal{Q}_\beta^{s,t}$ is *g-disconnected*, we propose to transfer the *logmap* and *expmap* into the diffeomorphic manifold $\psi : \mathcal{Q}_\beta^{s,t} \to \mathbb{S}_{-\beta}^t \times \mathbb{R}^s$, since the product manifold $\mathbb{S}_{-\beta}^t \times \mathbb{R}^s$ is *g-connected*. To map tangent vectors between tangent space of $\mathcal{Q}_\beta^{s,t}$ and $\mathbb{S}_{-\beta}^t \times \mathbb{R}^s$, we exploit *pushforward* that induce a linear approximation of smooth maps on tangent spaces.

**Definition 2** (Pushforward). *Suppose that $\psi : \mathcal{M} \to \mathcal{M}'$ is a smooth map, then the differential of $\psi$: $d\psi$ at point $\mathbf{x}$ is a linear map from the tangent space of $\mathcal{M}$ at $\mathbf{x}$ to the tangent space of $\mathcal{M}'$ at $\psi(\mathbf{x})$. Namely, $d\psi : \mathcal{T}_x \mathcal{M} \to \mathcal{T}_{\psi(x)} \mathcal{M}'$*

Intuitively, *pushforward* can be used to *push* tangent vectors on $\mathcal{T}_x \mathcal{Q}_\beta^{s,t}$ forward to tangent vectors on $\mathcal{T}_{\psi(x)} \mathbb{S}_{-\beta}^t \times \mathbb{R}^s$. Based on this, the new *logmap* and its inverse *expmap* can be defined by Eq. (1).

$$\widehat{\log}_{\mathcal{Q}_\beta^{s,t}}(\mathbf{x}) = \psi^{-1}(\log_{\mathbb{S}_{-\beta}^t \times \mathbb{R}^s}(\psi(\mathbf{x}))), \quad \widehat{\exp}_{\mathcal{Q}_\beta^{s,t}}(\boldsymbol{\xi}) = \psi^{-1}(\exp_{\mathbb{S}_{-\beta}^t \times \mathbb{R}^s}(\psi(\boldsymbol{\xi}))), \quad (1)$$

where $\psi(\cdot)$ is the spherical projection and $\psi^{-1}(\cdot)$ is the inverse. The mapping of tangent vectors is achieved by *pushforward* operations. The operations $\log_{\mathbb{S}_{-\beta}^t \times \mathbb{R}^s}(\cdot)$ and $\exp_{\mathbb{S}_{-\beta}^t \times \mathbb{R}^s}(\cdot)$ in the product manifolds can be defined as the concatenation of corresponding operations in different components.

$$\log_{\mathbb{S}_{-\beta}^t \times \mathbb{R}^s}(\mathbf{x}') = \log_{\mathbb{S}_{-\beta}^t}(\mathbf{x}_t') \,\|\, \log_{\mathbb{R}^s}(\mathbf{x}_s'), \quad \exp_{\mathbb{S}_{-\beta}^t \times \mathbb{R}^s}(\boldsymbol{\xi}) = \exp_{\mathbb{S}_{-\beta}^t}(\boldsymbol{\xi}_t) \,\|\, \exp_{\mathbb{R}^s}(\boldsymbol{\xi}_s), \quad (2)$$

where $\|$ denotes the concatenation, $\mathbf{x}' = \psi_{\mathbb{S}}(\mathbf{x})$ consists of spherical features $\mathbf{x}_t' \in \mathbb{S}_{-\beta}^t$ and Euclidean features $\mathbf{x}_s' \in \mathbb{R}^s$. $\boldsymbol{\xi}$ is the tangent vector induced by $\mathbf{x}$ on $\mathcal{Q}_\beta^{s,t}$.

We choose points where space dimension $\mathbf{s} = \mathbf{0}$ as the reference points due to the following property.

**Theorem 3.** *For any reference point* $\mathbf{x} = \begin{pmatrix} \mathbf{t} \\ \mathbf{s} \end{pmatrix} \in \mathcal{Q}_\beta^{s,t}$ *with space dimension* $\mathbf{s} = \mathbf{0}$*, the induced tangent space of* $\mathcal{Q}_\beta^{s,t}$ *is equal to the tangent space of its diffeomorphic manifold* $\mathbb{S}_{-\beta}^t \times \mathbb{R}^s$*, namely,* $\mathcal{T}_\mathbf{x}(\mathbb{S}_{-\beta}^t \times \mathbb{R}^s) = \mathcal{T}_\mathbf{x} \mathcal{Q}_\beta^{s,t}$*. (proof in the Appendix D.3).*

The intuition of proof is that if space dimension $\mathbf{s} = \mathbf{0}$, the pushforward (differential) function just influences time dimension, for which the mapping is just an identity function (see Appendix D.2). In this way, although we transfer *logmap* and *expmap* to the diffeomorphic manifold $\mathbb{S}_{-\beta}^t \times \mathbb{R}^s$, the diffeomorphic operations $\widehat{\log}_\mathbf{x}(\cdot)$ and $\widehat{\exp}_\mathbf{x}(\cdot)$ are still bijective functions from the pseudo-hyperboloid to the tangent space of the manifold itself. Hence, our final operations are actually still defined in the tangent space of the pseudo-hyperboloid. Note that such property only holds when our Theorem 2 is applied and the special reference points with space dimension $\mathbf{s} = \mathbf{0}$ are chosen.

By leveraging the new *logmap* and *expmap*, we further formulate the diffeomorphic version of tangential operations as follows.

**Tangential operations.** For function $f : \mathbb{R}^d \to \mathbb{R}^{d'}$, the pseudo-hyperboloid version $f^\otimes : \mathcal{Q}_\beta^{s,t} \to \mathcal{Q}_\beta^{s',t'}$ with $s + t = d$ and $s' + t' = d'$ can be defined by the means of $\widehat{\log}_\mathbf{x}^\beta(\cdot)$ and $\widehat{\exp}_\mathbf{x}^\beta(\cdot)$ as Eq. (3).

$$f^\otimes(\cdot) := \widehat{\exp}_\mathbf{x}^\beta \left( f \left( \widehat{\log}_\mathbf{x}^\beta (\cdot) \right) \right), \tag{3}$$

where $\mathbf{x}$ is the reference point. Note that this function is a morphism (i.e. $(f \circ g)^\otimes = f^\otimes \circ g^\otimes$) and direction preserving (i.e. $f^\otimes(\cdot)/\|f^\otimes(\cdot)\| = f(\cdot)/\|f(\cdot)\|$) [15], making it a natural way to define pseudo-hyperboloid version of vector operations such like scalar multiplication, matrix-vector multiplication, tangential aggregation and point-wise non-linearity and so on.

**Parallel transport.** Parallel transport is the generalization of Euclidean translation into manifolds. Formally, for any two points $\mathbf{x}$ and $\mathbf{y}$ connected by a geodesic, parallel transport $P_{\mathbf{x} \to \mathbf{y}}^\beta(\boldsymbol{\xi}) : \mathcal{T}_\mathbf{x}\mathcal{M} \to \mathcal{T}_\mathbf{y}\mathcal{M}$ is an isomorphism between two tangent spaces by moving one tangent vector $\boldsymbol{\zeta} \in \mathcal{T}_x\mathcal{M}$ with tangent direction $\boldsymbol{\xi} \in \mathcal{T}_x\mathcal{M}$ to another tangent space $\mathcal{T}_\mathbf{y}\mathcal{M}$. The parallel transport in pseudo-hyperboloid can be defined as the combination of Riemannian parallel transport [31]. However, the parallel transport has not been defined when there does not exist a geodesic between $\mathbf{x}$ and $\mathbf{y}$. i.e., the tangent vector $\boldsymbol{\zeta}$ induced by $\mathbf{x}$ can not be transported to the tangent space of points outside of the normal neighborhood $\mathcal{U}_\mathbf{x}$. Intuitively, the normal neighborhoods satisfy the following property.

**Theorem 4.** *For any point* $\mathbf{x} \in \mathcal{Q}_\beta^{s,t}$*, the union of the normal neighborhood of* $\mathbf{x}$ *and the normal neighborhood of its antipodal point* $-\mathbf{x}$ *cover the entire manifold. Namely,* $\mathcal{U}_\mathbf{x} \cup \mathcal{U}_{-\mathbf{x}} = \mathcal{Q}_\beta^{s,t}$ *(proof in the Appendix D.4).*

This theorem ensures that if a point $\mathbf{y} \notin \mathcal{U}_\mathbf{x}$, its antipodal point $-\mathbf{y} \in \mathcal{U}_\mathbf{x}$. Besides, $\mathcal{T}_\mathbf{y}\mathcal{M}$ is parallel to $\mathcal{T}_{-\mathbf{y}}\mathcal{M}$. Hence, $P_{\mathbf{x} \to \mathbf{y}}^\beta$ can be alternatively defined as $P_{\mathbf{x} \to -\mathbf{y}}^\beta$ for broken points. This result is crucial to define the pseudo-hyperbolic addition, such as bias translation, detailed in section 3.2.

**Broken geodesic distance.** By the means of geodesic, the induced distance between $\mathbf{x}$ and $\mathbf{y}$ in pseudo-hyperboloid is defined as the arc length of geodesic $\gamma(\tau)$, given by $\mathrm{d}_\gamma(\mathbf{x}, \mathbf{y}) = \sqrt{\|\log_\mathbf{x}(\mathbf{y})\|_t^2}$. For broken cases in which $\log_\mathbf{x}(\mathbf{y})$ is not defined, one approach is to use approximation like [24]. Different from that, we define following closed-form distance, given by Eq. (4).

$$\mathrm{D}_\gamma(\mathbf{x}, \mathbf{y}) = \begin{cases} \mathrm{d}_\gamma(\mathbf{x}, \mathbf{y}), & \text{if } \langle \mathbf{x}, \mathbf{y} \rangle_t < |\beta| \\ \pi\sqrt{|\beta|} + \mathrm{d}_\gamma(\mathbf{x}, -\mathbf{y}), & \text{if } \langle \mathbf{x}, \mathbf{y} \rangle_t \geq |\beta| \end{cases} \tag{4}$$

The intuition is that when $\mathbf{x}, \mathbf{y} \in \mathcal{Q}_\beta^{s,t}$ are *g-disconnected*, we consider the distance as $\mathrm{d}_\gamma(\mathbf{x}, \mathbf{y}) = \mathrm{d}_\gamma(\mathbf{x}, -\mathbf{x}) + \mathrm{d}_\gamma(-\mathbf{x}, \mathbf{y})$ or $\mathrm{d}_\gamma(\mathbf{x}, \mathbf{y}) = \mathrm{d}_\gamma(\mathbf{x}, -\mathbf{y}) + \mathrm{d}_\gamma(-\mathbf{y}, \mathbf{y})$. Since $\mathrm{d}_\gamma(\mathbf{x}, -\mathbf{x}) = \mathrm{d}_\gamma(-\mathbf{y}, \mathbf{y}) = \pi\sqrt{|\beta|}$ is a constant and $\mathrm{d}_\gamma(-\mathbf{x}, \mathbf{y}) = \mathrm{d}_\gamma(\mathbf{x}, -\mathbf{y})$, the distance between broken points can be calculated as $\mathrm{d}_\gamma(\mathbf{x}, \mathbf{y}) = \pi\sqrt{|\beta|} + \mathrm{d}_\gamma(\mathbf{x}, -\mathbf{y})$.

To clarify theoretical contributions, our Theorem 2 is nessasary for our Theorem 3 while Theorem 3 is nessasary for transforming the GCN operations directly into the tangent space of the pseudo-hyperboloid. Besides, we are the first to formulate the diffeomorphic *expmap*, *logmap* and tangential

operations of pseudo-hyperboloid to avoid broken cases. The theoretical properties of parallel transport and geodesic distance are discussed in the literature [24, 31]. However, we re-formulate parallel transport with Theorem 4 to avoid broken issues and propose a new distance measure using the broken geodesic (Eq.4), which is different from the approximated distance in [24].

## 3.2 Model architecture

GCNs can be interpreted as performing neighborhood aggregation after a linear transformation on node features of each layer. We present pseudo-Riemannian GCNs ($\mathcal{Q}$-GCN) by deriving corresponding operations with the developed geodesic tools in the $\mathcal{Q}_\beta^{s,t}$.

**Feature initialization.** We first map the features from Euclidean space to pseudo-hyperboloid, considering that the input features of nodes usually live in Euclidean space. Following the feature transformation from Euclidean space to pseudo-hyperboloid in [24], we initialize the node features by performing a differentiable mapping $\varphi : \mathbb{R}_*^{t+1} \times \mathbb{R}^s \to \mathcal{Q}_\beta^{s,t}$ that can be implemented by a double projection [24] based on Theorem 2, i.e. $\varphi = \psi^{-1} \circ \psi$. The intuition is that we first map the Euclidean features into diffeomorphic manifolds $\mathbb{S}_{-\beta}^t \times \mathbb{R}^s$ via $\psi(\cdot)$, and then map them into the pseudo-hyperboloid $\mathcal{Q}_\beta^{s,t}$ via $\psi^{-1}(\cdot)$, where the mapping functions are given by Eq. (5).

$$\psi(\mathbf{x}) = \begin{pmatrix} \sqrt{|\beta|}\frac{\mathbf{t}}{\|\mathbf{t}\|} \\ \mathbf{s} \end{pmatrix} \quad , \quad \psi^{-1}(\mathbf{z}) = \begin{pmatrix} \frac{\sqrt{|\beta|+\|\mathbf{v}\|^2}}{\sqrt{|\beta|}}\mathbf{u} \\ \mathbf{v} \end{pmatrix} , \tag{5}$$

where $\mathbf{x} = (\mathbf{t}, \mathbf{s})^\top \in \mathcal{Q}_\beta^{s,t}$ with $\mathbf{t} \in \mathbb{R}_*^t$ and $\mathbf{s} \in \mathbb{R}^s$. $\mathbf{z} = (\mathbf{u}, \mathbf{v})^\top \in \mathbb{S}_{-\beta}^t \times \mathbb{R}^s$ with $\mathbf{u} \in \mathbb{S}_{-\beta}^t$ and $\mathbf{v} \in \mathbb{R}^s$.

**Tangential aggregation.** The linear combination of neighborhood features is lifted to the tangent space, which is an intrinsic operation in differential manifolds [32, 16]. Specifically, $\mathcal{Q}$-GCN aggregates neighbours' embeddings in the tangent space of the reference point $\mathbf{o}$ before passing through a tangential activation function, and then projects the updated representation back to the manifold. Formally, at each layer $\ell$, the updated features of each node $i$ are defined as Eq. (6).

$$\mathbf{h}_i^{\ell+1} = \widehat{\exp}_{\mathbf{o}}^{\beta_{\ell+1}} \left( \sigma \left( \sum_{j \in \mathcal{N}(i) \cup \{i\}} \widehat{\log}_{\mathbf{o}}^{\beta_\ell} \left( W^\ell \otimes^{\beta_\ell} \mathbf{h}_j^\ell \oplus^{\beta_\ell} \mathbf{b}^\ell \right) \right) \right) , \tag{6}$$

where $\sigma(\cdot)$ is the activation function, $\beta_\ell$ and $\beta_{\ell+1}$ are two layer-wise curvatures, $\mathcal{N}(i)$ denotes the one-hop neighborhoods of node $i$, and the $\otimes, \oplus$ denote two basic operations, i.e. tangential transformation and bias translation, respectively.

**Tangential transformation.** We perform Euclidean transformations on the tangent space by leveraging the *expmap* and *logmap* in Eq. (1). Specifically, we first project the hidden feature into the tangent space of *south pole* $\mathbf{o} = [|\beta|, 0, , ..., 0]$ using *logmap* and then perform Euclidean matrix multiplication. Afterwards, the transformed features are mapped back to the manifold using *expmap*.

Formally, at each layer $\ell$, the tangential transformation is given by $W^\ell \otimes^\beta \mathbf{h}^\ell := \widehat{\exp}_{\mathbf{o}}^\beta (W^\ell \widehat{\log}_{\mathbf{o}}^\beta (\mathbf{h}^\ell))$, where $\otimes^\beta$ denotes the pseudo-hyperboloid tangential multiplication, and $W^\ell \in \mathbb{R}^{d' \times d}$ denotes the layer-wise learnable matrix in Euclidean space.

**Bias translation.** It is noteworthy that simply stacking multiple layers of the tangential transformation would collapse the composition [32, 15], i.e. $\exp_{\mathbf{o}}^\beta ...(W^1 \log_{\mathbf{o}}^\beta (\exp_{\mathbf{o}}^\beta (W^0 \log_{\mathbf{o}}^\beta (x)))) = \exp_{\mathbf{o}}^\beta (W^0 \times W^1 \times ... \times \log_{\mathbf{o}}^\beta (x))$, which means that these multiplications can simply be performed in Euclidean space except the first *logmap* and last *expmap*. To avoid model collapsing , we perform bias translation after the tangential transformation. By the means of pseudo-hyperboloid *parallel transport*, the bias translation can be performed by parallel transporting a tangent vector $\mathbf{b}^\ell \in \mathcal{T}_{\mathbf{o}} \mathcal{Q}_\beta^{s,t}$ to the tangent space of the point of interest. Finally, the transported tangent vector is mapped back to the manifold with *expmap*. Considering that $\widehat{\exp}(\cdot)$ is only defined at point $\mathbf{x} \in \mathcal{Q}_\beta^{s,t}$ with the space dimension $\mathbf{x}_s = \mathbf{0}$, we perform the original $\exp_{\mathcal{Q}_\beta^{s,t}}(\cdot)$ at the point of interest. The bias translation is formally given by:

$$\widetilde{\mathbf{h}}^\ell \oplus^\beta \mathbf{b}^\ell := \begin{cases} \exp_{\widetilde{\mathbf{h}}^\ell}^\beta \left( P_{\mathbf{o} \to \widetilde{\mathbf{h}}^\ell}^\beta (\mathbf{b}^\ell) \right), & \text{if } \langle \mathbf{o}, \widetilde{\mathbf{h}}^\ell \rangle_t < |\beta| \\ -\exp_{-\widetilde{\mathbf{h}}^\ell}^\beta \left( P_{\mathbf{o} \to -\widetilde{\mathbf{h}}^\ell}^\beta (\mathbf{b}^\ell) \right), & \text{if } \langle \mathbf{o}, \widetilde{\mathbf{h}}^\ell \rangle_t \geq |\beta| \end{cases} \tag{7}$$

where $\widetilde{\mathbf{h}}^\ell = W^\ell \otimes^\beta \mathbf{h}^\ell$, $\oplus^\beta$ denotes the pseudo-hyperboloid addition. For the broken cases where $\langle \mathbf{o}, \widetilde{\mathbf{h}}^\ell \rangle_t \geq |\beta|$, the parallel transport $P^\beta_{\mathbf{o} \to \widetilde{\mathbf{h}}^\ell}$ is not defined. In this case, we parallel transport $\mathbf{b}^\ell$ to the tangent space of the antipodal point $-\widetilde{\mathbf{h}}^\ell$, and then perform $\exp^\beta_{-\widetilde{\mathbf{h}}^\ell}$ to map it back to the manifold. Note that the case $\langle \mathbf{o}, \widetilde{\mathbf{h}}^\ell \rangle_t = |\beta|$ occurs if and only if $\mathbf{h} = -\mathbf{o}$, in which case $P^\beta_{\mathbf{o} \to -\widetilde{\mathbf{h}}^\ell}\left(\mathbf{b}^\ell\right) = P^\beta_{\mathbf{o} \to -\mathbf{o}}\left(\mathbf{b}^\ell\right) = -\mathbf{b}$.

### 3.3 Model training

Having introduced all the building blocks, $\mathcal{Q}$-GCN stacks multiple pseudo-Riemannian GCN layers and the final embeddings at the last layer can then be used to perform downstream tasks. For graph reconstruction, the objective is to map all nodes into a low-dimensional space such that the connected nodes are closer than unconnected nodes. Following [20, 24], we minimize the loss function $\mathcal{L}(\Theta) = \sum_{(u,v) \in \mathcal{D}} \log \frac{e^{-d(\boldsymbol{u},\boldsymbol{v})}}{\sum_{\boldsymbol{v}' \in \mathcal{E}(u)} e^{-d(\boldsymbol{u},\boldsymbol{v}')}}$ under the set of connected relations $\mathcal{D}$ in the graph, where $\mathcal{E}(u) = \{v | (u,v) \notin D, v \neq u\}$ is the set of negative examples for node $u$, $d(\cdot)$ is the distance function defined in Eq. (4). For node classification, we map the output of the last layer of $\mathcal{Q}$-GCN to the tangent space, and then perform Euclidean multinomial logistic regression. For link prediction, we utilize the Fermi-Dirac decoder [8] to compute probability scores for edges, formally given by $P(e_{uv} \in \mathcal{E} | \Theta) = \frac{1}{e^{(d(\mathbf{u},\mathbf{v})-r)/t}+1}$, where $r$ and $t$ are hyperparameters, $d(\mathbf{u}, \mathbf{v})$ is the distance function in the embedding space. We then train $\mathcal{Q}$-GCN by minimizing the cross-entropy loss using negative sampling.

**Optimization.** Although the model is built in $\mathcal{Q}^{s,t}_\beta$, the trainable parameters are all defined in Euclidean space through the diffeomorphic mappings. Following the standard tangential optimization strategy [32], the parameters can be optimized via Euclidean optimization by applying layer-wise diffeomorphic *expmap* and *logmap*. One optional strategy is to use dedicated optimization like [24], which we left for our future work.

**Complexity analysis.** The time complexity is the same as a vanilla GCN given by $O(|V|dd' + |E|d')$, where $|V|$ and $|E|$ are the number of nodes and edges, $d$ and $d'$ are the dimension of input and hidden features. The computation can be parallelized across all nodes. Similar to other non-Euclidean GCNs [32, 33, 17], the mapping from manifolds to the tangent space consume additional computation resources, compared with Euclidean GCNs, which is within the acceptable limits.

## 4 Experiments

We evaluate the effectiveness of $\mathcal{Q}$-GCN on graph reconstruction, node classification and link prediction. Firstly, we study the geometric properties of the used datasets including the graph sectional curvature [20] and the $\delta$-hyperbolicity [7]. Fig. 4 in the Appendix shows the histograms of sectional curvature and the mean sectional curvature for all datasets. It can be seen that all datasets have both positive and negative sectional curvatures, showcasing that all graphs contain mixed graph topologies. To further analyze the degree of the hierarchy, we apply $\delta$-hyperbolicity to identify the tree-likeness, as shown in Table 9 in the Appendix. We conjecture that the datasets with positive graph sectional curvature or larger $\delta$-hyperbolicity should be suitable for pseudo-hyperboloid with a smaller time dimension, while datasets with negative graph sectional curvature or smaller $\delta$-hyperbolicity should be aligned well with pseudo-hyperboloid with a larger time dimension.

### 4.1 Graph reconstruction

**Datasets and baselines.** We benchmark graph reconstruction on four real-world graphs including 1) Web-Edu [34]: a web network consisting of the $.edu$ domain; 2) Power [35]: a power grid distribution network with backbone structure; 3) Bio-Worm [36]: a worms gene network; 4) Facebook [37]: a dense social network from Facebook. We compare our method with Euclidean GCN [2], hyperbolic GCN (HGCN) [32], spherical GCN, and product manifold GCNs ($\kappa$-GCN) [33] with three signatures (i.e. $\mathbb{H}^5 \times \mathbb{H}^5$, $\mathbb{H}^5 \times \mathbb{S}^5$ and $\mathbb{S}^5 \times \mathbb{S}^5$). Besides, five variants of our model are implemented with different time dimension in $[1, 3, 5, 7, 10]$ for comparison.

Table 2: ROC AUC (%) for Link Prediction (LP) and F1 score for Node Classification (NC).

| Dataset | Airport | | Pubmed | | CiteSeer | | Cora | |
|---|---|---|---|---|---|---|---|---|
| $\delta$-hyperbolicity | 1.0 | | 3.5 | | 4.5 | | 11.0 | |
| Method | LP | NC | LP | NC | LP | NC | LP | NC |
| GCN | 89.24±0.21 | 81.54±0.60 | 91.31±1.68 | 79.30±0.60 | 85.48±1.75 | 72.27±0.64 | 88.52±0.85 | 81.90±0.41 |
| GAT | 90.35±0.30 | 81.55±0.53 | 87.45±0.00 | 78.30±0.00 | 87.24±0.00 | 71.10±0.00 | 85.73±0.01 | 83.05±0.08 |
| SAGE | 89.86±0.52 | 82.79±0.17 | 90.70±0.07 | 77.30±0.09 | 90.71±0.20 | 69.20±0.10 | 87.52±0.22 | 74.90±0.07 |
| SGC | 89.80±0.34 | 80.69±0.23 | 90.54±0.07 | 78.60±0.30 | 89.61±0.23 | 71.60±0.03 | 89.42±0.11 | 81.60±0.43 |
| HGCN ($\mathbb{H}^{16}$) | 96.03±0.26 | 90.57±0.36 | 96.08±0.21 | 80.50±1.23 | 96.31±0.41 | 68.90±0.63 | 91.62±0.33 | 79.90±0.18 |
| $\kappa$-GCN ($\mathbb{H}^{16}$) | 96.35±0.62 | 87.92±1.33 | 96.60±0.32 | 77.96±0.36 | 95.34±0.16 | 73.25±0.51 | 94.04±0.34 | 79.80±0.50 |
| $\kappa$-GCN ($\mathbb{S}^{16}$) | 90.38±0.32 | 81.94±0.58 | 94.84±0.13 | 78.80±0.49 | 95.79±0.24 | 72.13±0.51 | 93.20±0.48 | 81.08±1.45 |
| $\kappa$-GCN ($\mathbb{H}^{8} \times \mathbb{S}^{8}$) | 93.10±0.49 | 81.93±0.45 | 94.89±0.19 | 79.20±0.65 | 93.44±0.31 | 73.05±0.59 | 92.22±0.48 | 79.30±0.81 |
| $\mathcal{Q}$-GCN ($\mathcal{Q}^{15,1}$) | 96.30±0.22 | 89.72±0.52 | 95.42±0.22 | 80.50±0.26 | 94.76±1.49 | 72.67±0.76 | 93.14±0.30 | 80.57±0.20 |
| $\mathcal{Q}$-GCN ($\mathcal{Q}^{14,2}$) | 94.37±0.44 | 84.40±0.35 | 96.86±0.37 | 81.34±1.54 | 94.78±0.17 | 73.43±0.58 | 93.41±0.57 | 81.62±0.21 |
| $\mathcal{Q}$-GCN ($\mathcal{Q}^{13,3}$) | 92.53±0.17 | 82.38±1.53 | 96.20±0.34 | 80.94±0.45 | 94.54±0.16 | 74.13±1.41 | 93.56±0.18 | 79.91±0.42 |
| $\mathcal{Q}$-GCN ($\mathcal{Q}^{2,14}$) | 90.03±0.12 | 81.14±1.32 | 94.30±1.09 | 78.40±0.39 | 94.80±0.08 | 72.72±0.47 | 94.17±0.38 | 83.10±0.35 |
| $\mathcal{Q}$-GCN ($\mathcal{Q}^{1,15}$) | 89.07±0.58 | 81.24±0.34 | 94.66±0.18 | 78.11±1.38 | 97.01±0.30 | 73.19±1.58 | 94.81±0.27 | 83.72±0.43 |
| $\mathcal{Q}$-GCN ($\mathcal{Q}^{0,16}$) | 89.01±0.61 | 80.91±0.65 | 94.49±0.28 | 77.90±0.80 | 96.21±0.38 | 72.54±0.27 | 95.16±1.25 | 82.51±0.32 |

**Experimental settings.** Following [20, 24, 33], we use one-hot embeddings as initial node features. To avoid the time dimensions being $\mathbf{0}$, we uniformly perturb each dimension with a small random value in the interval $[-\epsilon, \epsilon]$, where $\epsilon = 0.02$ in practice. In addition, the same 10-dimensional embedding and 2 hidden layers are used for all baselines to ensure a fair comparison. The learning rate is set to 0.01, the learning rate of curvature is set to 0.0001. $\mathcal{Q}$-GCN is implemented with the Adam optimizer. We repeat the experiments 10 times via different random seeds influencing weight initialization and data batching.

Table 1: The graph reconstruction results in mAP (%), top three results are highlighted. Standard deviations are relatively small (in range $[0, 1.2 \times 10^{-2}]$) and are omitted.

| Model | Web-Edu | Power | Bio-Worm | Facebook |
|---|---|---|---|---|
| Curvature | -0.6 | -0.3 | 0.0 | 0.1 |
| GCN ($\mathbb{E}^{10}$) | 83.66 | 86.61 | 90.19 | 81.73 |
| HGCN ($\mathbb{H}^{10}$) | 88.33 | 93.80 | 93.12 | 83.40 |
| GCN ($\mathbb{S}^{10}$) | 82.72 | 92.73 | 88.98 | 81.04 |
| $\kappa$-GCN ($\mathbb{H}^{5} \times \mathbb{H}^{5}$) | 89.21 | 94.40 | 94.00 | 84.94 |
| $\kappa$-GCN ($\mathbb{S}^{5} \times \mathbb{S}^{5}$) | 86.70 | 94.58 | 90.36 | 84.56 |
| $\kappa$-GCN ($\mathbb{H}^{5} \times \mathbb{S}^{5}$) | 87.96 | 95.82 | 94.74 | 87.73 |
| $\mathcal{Q}$-GCN ($\mathcal{Q}^{9,1}$) | 87.03 | 94.35 | 92.83 | 81.60 |
| $\mathcal{Q}$-GCN ($\mathcal{Q}^{7,3}$) | 99.67 | 100.00 | 97.23 | 87.74 |
| $\mathcal{Q}$-GCN ($\mathcal{Q}^{5,5}$) | 98.49 | 100.00 | 95.75 | 87.03 |
| $\mathcal{Q}$-GCN ($\mathcal{Q}^{3,7}$) | 97.31 | 95.08 | 90.14 | 91.75 |
| $\mathcal{Q}$-GCN ($\mathcal{Q}^{0,10}$) | 82.57 | 94.20 | 88.67 | 83.81 |

**Results.** Table 1 shows the mean average precision (mAP) [20] results of graph reconstruction on four datasets. It shows that $\mathcal{Q}$-GCN achieves the best performance across all benchmarks compared with both Riemannian space and product manifolds. We observe that by setting proper signatures, the product spaces perform better than a single geometry. It is consistent with our statement that the expression power of a single view geometry is limited. Specifically, all the top three results are achieved by $\mathcal{Q}$-GCN, with one exception on Power where $\mathbb{H}^5 \times \mathbb{S}^5$ achieved the third-best performance. More precisely, for datasets that have smaller graph sectional curvature like Web-Edu, Power and Bio-Worm, $\mathcal{Q}^{7,3}$ perform the best, while $\mathcal{Q}^{3,7}$ perform the best on Facebook with positive sectional curvature. We conjecture that the number of time dimensions controls the geometry of the pseudo-hyperboloid. We find that the graphs with more hierarchical structures are inclined to be embedded with fewer time dimensions. By analyzing the sectional curvature in Fig. 4, we find that this makes sense as the mean sectional curvature of Power, Bio-Wormnet and Web-Edu are negative while it is negative for Facebook. Such results give us an intuition to determine the best time dimension based on the geometric properties of graphs.

## 4.2 Node classification and link prediction

**Datasets and baselines.** We consider four benchmark datasets: Airport, Pubmed, Citeseer and Cora, where Airport is airline networks, Pubmed, Citeseer and Cora are three citation networks. We observe that the graph sectional curvatures of the four datasets are consistently negative without significant differences in Fig. 4, hence we report the additional $\delta$-hyperbolicity for comparison in Table 2. GCN [2], GAT [3], SAGE [38] and SGC [4] are used as Euclidean GCN counterparts. For non-Euclidean GCN baselines, we compare HGCN [32] and $\kappa$-GCN [33] with its three variants as

Table 3: F1 score for node classification of MLP, HNN and $\mathcal{Q}$-NN on Pubmed, Citeseer and Cora.

| Method | Pubmed | CiteSeer | Cora |
|--------|--------|----------|------|
| MLP | 72.30±0.30 | 60.22±0.42 | 55.80±0.08 |
| HNN | 74.60±0.40 | 59.92±0.87 | 59.60±0.09 |
| $\mathcal{Q}$-NN ($\mathcal{Q}^{15,1}$) | 74.31±0.33 | 59.33±0.35 | 60.38±0.56 |
| $\mathcal{Q}$-NN ($\mathcal{Q}^{14,2}$) | **76.26±0.31** | **64.33±0.35** | 62.77±0.30 |
| $\mathcal{Q}$-NN ($\mathcal{Q}^{13,3}$) | 75.85±0.79 | 63.65±0.57 | 59.04±0.45 |
| $\mathcal{Q}$-NN ($\mathcal{Q}^{2,14}$) | 74.44±0.68 | 60.48±0.29 | 63.85±0.22 |
| $\mathcal{Q}$-NN ($\mathcal{Q}^{1,15}$) | 73.44±0.28 | 60.33±0.40 | **64.85±0.24** |
| $\mathcal{Q}$-NN ($\mathcal{Q}^{0,16}$) | 73.31±0.17 | 61.05±0.22 | 63.96±0.41 |

Table 4: The running time (sec) of graph reconstruction on Web-Edu and Facebook.

| Manifolds | Web-Edu | Facebook |
|-----------|---------|----------|
| GCN ($\mathbb{E}^{10}$) | 2284 | 5456 |
| Prod-GCN ($\mathbb{H}^5 \times \mathbb{S}^5$) | 4336 | 10338 |
| $\mathcal{Q}$-GCN ($\mathcal{Q}^{9,1}$) | 2769 | 6981 |
| $\mathcal{Q}$-GCN ($\mathcal{Q}^{7,3}$) | 3363 | 6303 |
| $\mathcal{Q}$-GCN ($\mathcal{Q}^{5,5}$) | 3620 | 7142 |
| $\mathcal{Q}$-GCN ($\mathcal{Q}^{3,7}$) | 3685 | 7512 |
| $\mathcal{Q}$-GCN ($\mathcal{Q}^{1,9}$) | 3532 | 7980 |
| $\mathcal{Q}$-GCN ($\mathcal{Q}^{0,10}$) | 2778 | 7037 |

explained before. For $\mathcal{Q}$-GCN, we empirically set the time dimension as $[1, 2, 3, 14, 15, 16]$ as six variants since these settings best reflect the geometric properties of hyperbolic and spherical space, respectively.

**Experimental settings.** For node classification, we use the same dataset split as [39] for citation datasets, where 20 nodes per class are used for training, and 500 nodes are used for validation and 1000 nodes are used for testing. For Airport, we split the dataset into $70/15/15$. For link prediction, the edges are split into $85/5/10$ percent for training, validation and testing for all datasets. To ensure a fair comparison, we set the same 16-dimension hidden embedding, 0.01 initial learning rate and 0.0001 learning rate for curvature. The optimal regularization with weight decay, dropout rate, the number of layers and activation functions are obtained by grid search for each method. We report the mean accuracy over 10 random seeds influencing weight initialization and batching sequence.

**Results.** Table 2 shows the averaged ROC AUC for link prediction, and F1 score for node classification. As we can see from the $\delta$-hyperbolicity, Airport and Pubmed are more hierarchical than CiteSeer and Cora. For Airport and Pubmed with dominating hierarchical properties (lower $\delta$), $\mathcal{Q}$-GCNs with fewer time dimensions achieve the results on par with hyperbolic space based methods such as HGCN [32], $\kappa$-GCN ($\mathbb{H}^{16}$). While for CiteSeer and Cora with less tree-like properties (higher $\delta$), $\mathcal{Q}$-GCNs achieve the state-of-the-art results, showcasing the flexibility of our model to embed complex graphs with different curvatures. More specifically, $\mathcal{Q}$-GCN with more time dimensions consistently performs best on Cora. While for CiteSeer, albeit $\mathcal{Q}$-GCN achieves the best results, the corresponding best variants are not consistent on both tasks.

### 4.3 Parameter sensitivity and analysis

**Time dimension.** We study the influence of time dimension for graph reconstruction by setting varying number of time dimensions under the condition of $s + t = 10$. Fig. 2 shows that the *time dimension* $t$ acts as a knob for controlling the geometric properties of $\mathcal{Q}_\beta^{s,t}$. The best performance are achieved by neither hyperboloid ($t = 1$) nor sphere cases ($t = 10$), showcasing the advantages of $\mathcal{Q}_\beta^{s,t}$ on representing graphs of mixed topologies. It shows that on Web-Edu, Power and Bio-Worm with smaller mean sectional curvature, after the optimal value is reached at a lower $t$, the performance decrease as $t$ increases. While on Facebook with larger (positive) mean sectional curvature, as $t$ rises, the effect gradually increases until it reaches a peak at a higher $t$. It is consistent with our hypothesis that graphs with

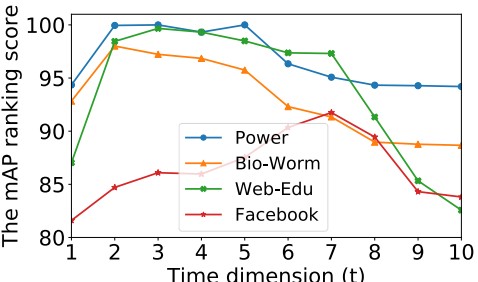

Figure 2: The mAP of graph reconstruction with varying number of time dimensions.

more hierarchical structure are inclined to be embedded in $\mathcal{Q}_\beta^{s,t}$ with smaller $t$, while cyclical data is aligned well with larger $t$. The results give us an intuition to determine the best time dimension based on the geometric properties of graphs.

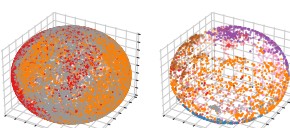 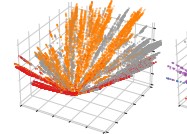 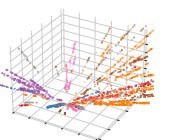 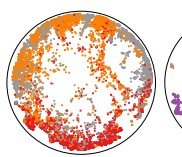 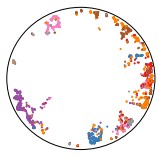

(a) Spherical projection      (b) Hyperbolic projection (3D)      (c) Hyperbolic projection (disk)

Figure 3: Visualization of the learned embeddings for link prediction on Pubmed (left) and Cora (right), where the colors denote the class of nodes. We apply (a) spherical projection, (b) hyperbolic projection (3D) and (c) hyperbolic projection (Poincaré disk) on the learned embeddings of $\mathcal{Q}$-GCN to visualize various views of the learned embeddings.

$\mathcal{Q}$-**NN VS** $\mathcal{Q}$-**GCN.** We also introduce $\mathcal{Q}$-NN, a generalization of MLP into pseudo-Riemannian manifold, defined as multiple layers of $f(\mathbf{x}) = \sigma^{\otimes}(W \otimes^{\beta} \mathbf{x} \oplus^{\beta} \mathbf{b})$, where $\sigma^{\otimes}$ is the tangential activation. Table 3 shows that $\mathcal{Q}$-NN with appropriate time dimension outperforms MLP and HNN on node classification, showcasing the expression power of pseudo-hyperboloid. Furthermore, compared with the results of $\mathcal{Q}$-GCN in Table 2, $\mathcal{Q}$-GCN performs better than $\mathcal{Q}$-NN, suggesting that the benefits of the neighborhood aggregation equipped with the proposed GCN operations.

**Computation efficiency.** We compare the running time of $\mathcal{Q}$-GCN, GCN and Prod-GCN per epoch. Table 4 shows that $\mathcal{Q}$-GCN achieves higher efficiency than Prod-GCN ($\mathbb{H}^5 \times \mathbb{S}^5$). This is mainly owing to that the component $\mathbb{R}$ in our diffeomorphic manifold ($\mathbb{S} \times \mathbb{R}$) runs faster than non-Euclidean components in $\mathbb{H}^5 \times \mathbb{S}^5$. The additional running time mainly comes from the mapping operations and the projection from the time dimensions to $\mathbb{S}$. The running time grows when increasing the number of time dimensions. Overall, albeit slower than Euclidean GCN, the running time of all variants of $\mathcal{Q}$-GCN is smaller than the twice of time in Euclidean GCN, which is within the acceptable limits.

**Visualization.** To visualize the embeddings learned by $\mathcal{Q}$-GCN, we use UMAP tool [2] to project the learned embeddings for Link Prediction on Pubmed and Cora into low-dimensional spherical and hyperbolic spaces. The projections include spherical projection into 3D sphere, hyperbolic projection into 3D plane, and 2D Poincaré disk. As shown in Fig. 3 (a,b), for Pubmed with more hyperbolic structures, the class separability is more significant in hyperbolic projection than that is in spherical projection. While the corresponding result is opposite for less tree-like Cora. Furthermore, Fig. 3 (c) provides a more clear insight of the hierarchy. It shows that there are more hub nodes near the origin of Poincaré disk in Pubmed than in Cora, showcasing the dominating tree-likeness of Pubmed.

## 5 Conclusion

In this paper, we generalize GCNs to pseudo-Riemannian manifolds of constant nonzero curvature with elegant theories of diffeomorphic geometry tools. The proposed $\mathcal{Q}$-GCN have the flexibility to fit complex graphs with mixed curvatures and have shown promising results on graph reconstruction, node classification and link prediction. One limitation might be the choice of time dimension, we provide some insights to decide the best time dimension but this could still be improved, which we left for our future work. The developed geodesic tools are application-agnostic and could be extended to more deep learning methods. We foresee our work would shed light on the direction of non-Euclidean geometric deep learning.

## Acknowledgments

The authors thank the International Max Planck Research School for Intelligent Systems (IMPRS-IS) for supporting Bo Xiong. Bo Xiong was funded by the European Union's Horizon 2020 research and innovation programme under the Marie Skłodowska-Curie grant agreement No: 860801. Nico Potyka was partially funded by DFG projects Evowipe/COFFEE. Shirui Pan was supported in part by an ARC Future Fellowship (FT210100097). This work was funded by Deutsche Forschungsgemeinschaft (DFG, German Research Foundation) under Germanys' Excellence Strategy—EXC 2075-390740016 (SimTech).

---

[2]https://umap-learn.readthedocs.io/en/latest/index.html

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
