# Appendices

## Broader Impact

Going beyond Riemannian manifolds, we extend GCNs to the pseudo-Riemannian manifolds, which is a smooth manifold furnished with indefinite metric. Our contribution is both theoretical and practical. We introduce necessary geodesic tools in geodesically disconnected pseudo-Riemannian manifolds to help machine learning researchers to define vector operations like matrix multiplication and vector addition. These operations are application agnostic and could be applied in the context of machine learning algorithms. $\mathcal{Q}$-GCN is a flexible framework and is especially powerful on modeling real-world graphs such as social networks and molecular networks which usually exhibit heterogeneous topological structures. We foresee our method can be applied to more practical settings such as recommender systems and drug discovery, and also stimulate more applications of pseudo-Riemannian geometry in machine learning and related communities. However, we emphasize that current models might suffer from the same problems as other deep models such as limited interpretability, and thus, are not capable of replacing human expertise. We advocate researchers to focus more on the interpretability of non-Euclidean graph embeddings.

## A   Notation

In this paper, we denote the points on manifolds by boldface Roman characters. The tangent vectors on tangent spaces are denoted by boldface Greek fonts. The notations are summarized in Table 5.

Table 5: Commonly used notations.

| Notations | Descriptions |
|---|---|
| $\mathcal{Q}_\beta^{s,t}$ | The pseudo-hyperboloid with $s$ space dimensions, $t$ time dimensions, and curvature parameter $\beta$ |
| $\mathbf{x}, \mathbf{y}, \mathbf{z}$ | The points in manifolds |
| $\boldsymbol{\xi}, \boldsymbol{\zeta}$ | The tangent vectors in tangent space |
| $\mathcal{T}_x \mathcal{M}$ | The tangent space at point $\mathbf{x} \in \mathcal{M}$ |
| $\mathbb{R}$ | The Euclidean space |
| $\mathbb{S}$ | The spherical space |
| $\mathbb{H}$ | The hyperbolic space |
| $\mathcal{U}_\mathbf{x}$ | The normal neighborhood of point $\mathbf{x}$ |
| $\gamma_{\mathbf{x} \to \boldsymbol{\xi}}(\tau)$ | The geodesic of mapping a real value $\tau$ to a point on $\mathcal{M}$ with tangent direction $\boldsymbol{\xi} \in \mathcal{T}_\mathbf{x} \mathcal{M}$ |
| $\exp, \log$ | The exponential and logarithmic maps |
| $\widehat{\exp}, \widehat{\log}$ | The diffeomorphic exponential and logarithmic maps |
| $P_{\mathbf{x} \to \mathbf{y}}^\beta(\boldsymbol{\xi})$ | The parallel transport of tangent vector on $\mathcal{T}_\mathbf{x} \mathcal{M}$ with direction $\boldsymbol{\xi} \in \mathcal{T}_\mathbf{x} \mathcal{M}$ to another tangent space $\mathcal{T}_\mathbf{y} \mathcal{M}$ |
| $\psi$ | The diffeomorphic function |
| $\otimes^\beta$ | Pseudo-hyperboloid multiplication |
| $\oplus^\beta$ | Pseudo-hyperboloid addition |

## B   Related Works

**Riemannian embedding.** Non-Euclidean Riemannian spaces have recently gained extensive attention in learning representation for non-euclidean data. [40] was the first work to learn hierarchical embeddings in hyperbolic space for link prediction. Following this work, [41] applied hyperbolic embedding in word embedding. [42] proposed a translation model in hyperbolic space for multi-relational graph embedding, and [43] proposed a hyperbolic embedding for answering complex logical queries. On the other hand, spherical space offers benefits for embedding spherical or cyclical data [14, 44, 45]. For mixed curvature space, product manifolds [20, 21] on Euclidean, hyperbolic

and spherical space has been proposed to embed graph and knowledge graphs [46] with mixed structures. HNN [15] extended neural networks into the hyperbolic space and defined some fundamental neural operations.

**Riemannian GCN.** Going beyond GCNs in Euclidean space [1, 2], HGNN [16] and HGCN [32] first extended GCNs into hyperbolic space, achieving state-of-the-art performance on learning graph embedding for scale-free networks. The main solution is to move the graph convolutional operations (e.g. *feature transformation*, *aggregation*) into the *tangent space* of manifolds and perform Euclidean operations on the space. To model graphs of mixed topologies, $\kappa$-GCN [33] unifies curvatures in a $\kappa$-stereographic model and extends GCNs into the products of Riemannian projection manifolds. GIL [17] proposed graph geometric interaction learning that models interaction between hyperbolic and Euclidean space. Different from these works, $\mathcal{Q}$-GCN learns graph embeddings in a single pseudo-Riemannian space with indefinite metrics while capturing mixed graph structures.

**Pseudo-Riemannian machine learning.** Few works [5, 31, 24, 25, 23] have explored the application of pseudo-Riemannian geometry in machine learning. [5] first applied pseudo-Riemannian geometry (or *pseudo-Euclidean space*) to embed non-metric data that preserves local information. [23] exploited Lorentzian spacetime on embedding directed acyclic graphs, [31] developed a manifold learning framework on the pseudo-Riemannian manifold and its submanifolds. More recently, [24] proposed learning graph embeddings on pseudo-hyperboloid and provided some geodesic tools. [25] further extended it into directed graph embedding with novel tools. [47] first studied the pseudo-Riemannian geometry in the setting of deep learning. However, their embedding is learned in a quotient manifold. Different from [47], our embedding is directly learned from the pseudo-hyperboloid.

## C   Review of Pseudo-Riemannian Geometry

### C.1   Pseudo-Riemannian manifold

The pseudo-Riemannian manifold $\mathcal{M}$ is equipped with a pseudo-Riemannian metric $g_{\mathbf{x}} : \mathcal{T}_{\mathbf{x}}\mathcal{M} \times \mathcal{T}_{\mathbf{x}}\mathcal{M}$ at point $\mathbf{x}$. The metric $g_{\mathbf{x}}$ is a nondegenerate symmetric bilinear form, which means that if for a given $\boldsymbol{\xi} \in \mathcal{T}_{\mathbf{x}}\mathcal{M}$, for any $\boldsymbol{\zeta} \in \mathcal{T}_{\mathbf{x}}\mathcal{M}$ we have $g_{\mathbf{x}}(\boldsymbol{\xi}, \boldsymbol{\zeta}) = 0$, then $\boldsymbol{\xi} = \mathbf{0}$. The $\mathcal{M}$ manifold has the constant sectional curvature of $1/\beta$ and constant mean curvature $\kappa = |\beta|^{-1/2}$. We recommend the reader to refer [27, 48] for details.

### C.2   Geodesic tools of pseudo-hyperboloid

Although pseudo-hyperboloid is a *geodescially disconnected* manifold, previous works [24, 31] have defined some standard geodesic tools that do not consider the broken cases which we recall below.

**Geodesic.** The geodesics of pseudo-hyperboloid are a combination of the hyperbolic, flat and spherical cases, depending on the sign of $\langle \boldsymbol{\xi}, \boldsymbol{\xi} \rangle_t$. Formally, the geodesic $\gamma_{\mathbf{x} \to \boldsymbol{\xi}}(\tau)$ of $\mathcal{Q}_{\beta}^{s,t}$ with $\beta < 0$ is defined as Eq. (8).

$$
\gamma_{\mathbf{x} \to \boldsymbol{\xi}}(\tau) = \begin{cases} \cosh\left(\frac{\tau\sqrt{|\langle\boldsymbol{\xi},\boldsymbol{\xi}\rangle_t|}}{\sqrt{|\beta|}}\right)\mathbf{x} + \frac{\sqrt{|\beta|}}{\sqrt{\langle\boldsymbol{\xi},\boldsymbol{\xi}\rangle_t}}\sinh\left(\frac{\tau\sqrt{|\langle\boldsymbol{\xi},\boldsymbol{\xi}\rangle_t|}}{\sqrt{|\beta|}}\right), & \text{if } \langle\boldsymbol{\xi},\boldsymbol{\xi}\rangle_t > 0 \\ \mathbf{x} + \tau\boldsymbol{\xi}, & \text{if } \langle\boldsymbol{\xi},\boldsymbol{\xi}\rangle_t = 0 \\ \cos\left(\frac{\tau\sqrt{|\langle\boldsymbol{\xi},\boldsymbol{\xi}\rangle_t|}}{\sqrt{|\beta|}}\right)\mathbf{x} + \frac{\sqrt{|\beta|}}{\sqrt{\langle\boldsymbol{\xi},\boldsymbol{\xi}\rangle_t}}\sin\left(\frac{\tau\sqrt{|\langle\boldsymbol{\xi},\boldsymbol{\xi}\rangle_t|}}{\sqrt{|\beta|}}\right) & \text{if } \langle\boldsymbol{\xi},\boldsymbol{\xi}\rangle_t < 0 \end{cases} \quad (8)
$$

**Exponential and logarithmic maps.** The exponential and logarithmic maps in the pseudo-hyperboloid manifold are defined as $\exp_{\mathbf{x}} : \mathcal{T}_{\mathbf{x}}\mathcal{Q}_{\beta}^{s,t} \to \mathcal{Q}_{\beta}^{s,t}$ and $\log_{\mathbf{x}} : \mathcal{Q}_{\beta}^{s,t} \to \mathcal{T}_{\mathbf{x}}\mathcal{Q}_{\beta}^{s,t}$, respectively. We have the following closed form expressions of the exponential and the logarithmic maps [24], which allow us to perform operations on points on the pseudo-hyperboloid or pseudo-sphere manifolds by mapping them to *tangent* spaces and vice-versa, formally given by

$$
\exp_{\mathbf{x}}(\boldsymbol{\xi}) = \begin{cases} \cosh\left(\frac{\sqrt{\langle\boldsymbol{\xi},\boldsymbol{\xi}\rangle}}{\sqrt{|\beta|}}\right)\mathbf{x} + \frac{\sqrt{|\beta|}}{\sqrt{\langle\boldsymbol{\xi},\boldsymbol{\xi}\rangle}}\sinh\left(\frac{t\sqrt{\langle\boldsymbol{\xi},\boldsymbol{\xi}\rangle}}{\sqrt{|\beta|}}\right)\boldsymbol{\xi}, & \text{if}\langle\boldsymbol{\xi},\boldsymbol{\xi}\rangle > 0 \\ \mathbf{x} + \boldsymbol{\xi}, & \text{if } \langle\boldsymbol{\xi},\boldsymbol{\xi}\rangle = 0 \\ \cos\left(\frac{\sqrt{\langle\boldsymbol{\xi},\boldsymbol{\xi}\rangle}}{\sqrt{|\beta|}}\right)\mathbf{x} + \frac{\sqrt{|\beta|}}{\sqrt{\langle\boldsymbol{\xi},\boldsymbol{\xi}\rangle}}\sin\left(\frac{\sqrt{\langle\boldsymbol{\xi},\boldsymbol{\xi}\rangle}}{\sqrt{|\beta|}}\right)\boldsymbol{\xi}, & \text{if } \langle\boldsymbol{\xi},\boldsymbol{\xi}\rangle < 0 \end{cases} \quad (9)
$$

$$\log_{\mathbf{x}}(\mathbf{y}) = \begin{cases} \frac{\cosh^{-1}\left(\frac{\langle x,y \rangle_t}{\beta}\right)}{\sqrt{\left(\frac{\langle x,y \rangle_t}{\beta}\right)^2 - 1}}\left(\mathbf{y} - \frac{\langle \mathbf{x},\mathbf{y} \rangle_t}{\beta}\mathbf{x}\right), & \text{if } \frac{\langle \mathbf{x},\mathbf{y} \rangle_t}{|\beta|} < -1 \\ \mathbf{y} - \mathbf{x}, & \text{if } \frac{\langle \mathbf{x},\mathbf{y} \rangle_t}{|\beta|} = -1 \\ \frac{\cos^{-1}\left(\frac{\langle \mathbf{x},\mathbf{y} \rangle_t}{\beta}\right)}{\sqrt{1 - \left(\frac{\langle \mathbf{x},\mathbf{y} \rangle_t}{\beta}\right)^2}}\left(\mathbf{y} - \frac{\langle \mathbf{x},\mathbf{y} \rangle_t}{\beta}\mathbf{x}\right), & \text{if } \frac{\langle \mathbf{x},\mathbf{y} \rangle_t}{|\beta|} \in (-1,1) \end{cases} \tag{10}$$

It is worth mentioning that not all points on the pseudo-hyperboloid manifold are connected by a geodesic. For these broken points, there does not exist a tangent vector $\boldsymbol{\xi}$ such that $\mathbf{y} = \exp_{\mathbf{x}}(\boldsymbol{\xi})$. Namely, the logarithmic map is not defined if $\langle \mathbf{x}, \mathbf{y} \rangle \geq |\beta|$.

**Distance.** By the means of the geodesic, the distance between $\mathbf{x}$ and $\mathbf{y}$ in pseudo-Riemannian manifolds $\mathcal{Q}_\beta^{s,t}$ is defined as the arc length of geodesic $\gamma(\tau)$, given by Eq. (11).

$$d(\mathbf{x}, \mathbf{y}) = \sqrt{\left| \|\log_{\mathbf{x}}(\mathbf{y})\|_t^2 \right|}. \tag{11}$$

**Parallel transport.** Given the geodesic $\gamma(\tau)$ on $\mathcal{Q}_\beta^{s,t}$ passing through $\mathbf{x} \in \mathcal{Q}_\beta^{s,t}$ with the tangent direction $\boldsymbol{\xi} \in \mathcal{T}_{\mathbf{x}}\mathcal{Q}_\beta^{s,t}$, the parallel transport of $\boldsymbol{\zeta} \in \mathcal{T}_{\mathbf{x}}\mathcal{Q}_\beta^{s,t}$ is defined as Eq. (12).

$$P_{\mathbf{x} \to \mathbf{y}}^\beta(\boldsymbol{\xi}) = \begin{cases} \frac{\langle \boldsymbol{\zeta},\boldsymbol{\xi} \rangle}{\|\boldsymbol{\xi}\|}\left[\mathbf{x}\sinh(\tau\|\boldsymbol{\xi}\|) + \frac{\boldsymbol{\xi}}{\|\boldsymbol{\xi}\|}\cosh(\tau\|\boldsymbol{\xi}\|)\right] + \left(\boldsymbol{\zeta} - \frac{\langle \boldsymbol{\zeta},\boldsymbol{\xi} \rangle}{\|\boldsymbol{\xi}\|^2}\boldsymbol{\xi}\right), & \text{if } \langle \boldsymbol{\xi},\boldsymbol{\xi} \rangle > 0 \\ \frac{\langle \boldsymbol{\zeta},\boldsymbol{\xi} \rangle}{\|\boldsymbol{\xi}\|}\left[\mathbf{x}\sin(\tau\|\boldsymbol{\xi}\|) - \frac{\boldsymbol{\xi}}{\|\boldsymbol{\xi}\|}\cos(\tau\|\boldsymbol{\xi}\|)\right] + \left(\boldsymbol{\zeta} + \frac{\langle \boldsymbol{\zeta},\boldsymbol{\xi} \rangle}{\|\boldsymbol{\xi}\|^2}\boldsymbol{\xi}\right), & \text{if } \langle \boldsymbol{\xi},\boldsymbol{\xi} \rangle < 0 \\ \langle \boldsymbol{\zeta},\boldsymbol{\xi} \rangle\left(\tau\mathbf{x} + \frac{1}{2}\tau^2\boldsymbol{\xi}\right) + \boldsymbol{\zeta}, & otherwise \end{cases} \tag{12}$$

where $\boldsymbol{\xi} = \log_{\mathbf{x}}(\mathbf{y})$.

# D Proof of Theorems

## D.1 Proof of Theorem 1

**Theorem 1** (Theorem 4.1 in [24]). *For any point $\mathbf{x} \in \mathcal{Q}_\beta^{s,t}$, there exists a diffeomorphism $\psi : \mathcal{Q}_\beta^{s,t} \to \mathbb{S}_1^t \times \mathbb{R}^s$ that maps $\mathbf{x}$ into the product manifolds of an unit sphere and the Euclidean space, the mapping and its inverse are given by,*

$$\psi(\mathbf{x}) = \begin{pmatrix} \frac{1}{\|t\|}\mathbf{t} \\ \frac{1}{\sqrt{|\beta|}}\mathbf{s} \end{pmatrix}, \quad \psi^{-1}(\mathbf{z}) = \sqrt{|\beta|}\begin{pmatrix} \sqrt{1 + \|\mathbf{v}\|^2}\mathbf{u} \\ \mathbf{v} \end{pmatrix}, \tag{13}$$

*where $\mathbf{x} = \begin{pmatrix} \mathbf{t} \\ \mathbf{s} \end{pmatrix} \in \mathcal{Q}_\beta^{s,t}$ with $\mathbf{t} \in \mathbb{R}_*^{t+1}$ and $\mathbf{s} \in \mathbb{R}^s$. $\mathbf{z} = \begin{pmatrix} \mathbf{u} \\ \mathbf{v} \end{pmatrix} \in \mathbb{S}_1^t \times \mathbb{R}^s$ with $\mathbf{u} \in \mathbb{S}_1^t$ and $\mathbf{v} \in \mathbb{R}^s$.*

Please refer Appendix C.5 in [24] for proof of this theorem.

## D.2 Proof of Theorem 2

**Theorem 2.** *For any point $\mathbf{x} \in \mathcal{Q}_\beta^{s,t}$, there exists a diffeomorphism $\psi : \mathcal{Q}_\beta^{s,t} \to \mathbb{S}_{-\beta}^t \times \mathbb{R}^s$ that maps $\mathbf{x}$ into the product manifolds of a sphere and the Euclidean space, the mapping and its inverse are given by,*

$$\psi(\mathbf{x}) = \begin{pmatrix} \sqrt{|\beta|}\frac{\mathbf{t}}{\|\mathbf{t}\|} \\ \mathbf{s} \end{pmatrix}, \quad \psi^{-1}(\mathbf{z}) = \begin{pmatrix} \frac{\sqrt{|\beta| + \|\mathbf{v}\|^2}}{\sqrt{|\beta|}}\mathbf{u} \\ \mathbf{v} \end{pmatrix}, \tag{14}$$

*where $\mathbf{x} = \begin{pmatrix} \mathbf{t} \\ \mathbf{s} \end{pmatrix} \in \mathcal{Q}_\beta^{s,t}$ with $\mathbf{t} \in \mathbb{R}_*^t$ and $\mathbf{s} \in \mathbb{R}^s$. $\mathbf{z} = \begin{pmatrix} \mathbf{u} \\ \mathbf{v} \end{pmatrix} \in \mathbb{S}_{-\beta}^t \times \mathbb{R}^s$ with $\mathbf{u} \in \mathbb{S}_{-\beta}^t$ and $\mathbf{v} \in \mathbb{R}^s$.*

*Proof.* It is easy to show that the $\psi$ and $\psi^{-1}$ are smooth functions as they only involve with a linear mapping with a constant scaling vector. Hence, we only need to show that $\psi\left(\psi^{-1}(\mathbf{z})\right) = \mathbf{z}$ and $\psi^{-1}(\psi(\mathbf{x})) = \mathbf{x}$. Here, we consider space dimensions and time dimensions separately.

For space dimensions, the mapping of the space dimensions of $\mathbf{x}$ to the space dimensions of $\mathbf{z}$ is an identity function (i.e. $\mathbf{v} = \mathbf{s}, \mathbf{s} = \mathbf{v}$). Thus, we only need to show the invertibility of the mappings taking time dimensions as inputs. For time dimensions, we first show that:

$$\psi^{-1}(\psi(\mathbf{t})) = \frac{\sqrt{|\beta| + \|\mathbf{v}\|^2}}{\sqrt{|\beta|}} \sqrt{|\beta|} \frac{\mathbf{t}}{\|\mathbf{t}\|} = \sqrt{|\beta| + \|\mathbf{v}\|^2} \frac{\mathbf{t}}{\|\mathbf{t}\|}$$

$$= \sqrt{|\beta| + \|\mathbf{s}\|^2} \frac{\mathbf{t}}{\|\mathbf{t}\|} = \|\mathbf{t}\| \frac{\mathbf{t}}{\|\mathbf{t}\|} = \mathbf{t}. \tag{15}$$

Note that the last equality can be inferred by the fact that $\mathbf{x} \in \mathcal{Q}_\beta^{s,t}$ and $\beta < 0$. We then show that:

$$\psi(\psi^{-1}(\mathbf{u})) = \sqrt{|\beta|} \frac{\sqrt{|\beta| + \|\mathbf{v}\|^2}}{\sqrt{|\beta|}} \frac{\mathbf{u}}{\|\frac{\sqrt{|\beta| + \|\mathbf{v}\|^2}}{\sqrt{|\beta|}} \mathbf{u}\|} = \sqrt{|\beta|} \frac{\mathbf{u}}{\|\mathbf{u}\|} = \mathbf{u}. \tag{16}$$

Note that the last equality can be inferred by the fact that $\mathbf{u} \in \mathbb{S}_{-\beta}^t$ and $\beta < 0$. $\square$

### D.3 Proof of Theorem 3

**Theorem 3.** *For any reference point* $\mathbf{x} = \begin{pmatrix} \mathbf{t} \\ \mathbf{s} \end{pmatrix} \in \mathcal{Q}_\beta^{s,t}$ *with space dimension* $\mathbf{s} = \mathbf{0}$*, the induced tangent space of* $\mathcal{Q}_\beta^{s,t}$ *is equal to the tangent space of its diffeomorphic manifold* $\mathbb{S}_{-\beta}^t \times \mathbb{R}^s$*, namely,* $\mathcal{T}_{\psi(\mathbf{x})}(\mathbb{S}_{-\beta}^t \times \mathbb{R}^s) = \mathcal{T}_\mathbf{x}\mathcal{Q}_\beta^{s,t}$.

*Proof.* For any point $\mathbf{x} = \begin{pmatrix} \mathbf{t} \\ \mathbf{s} \end{pmatrix} \in \mathcal{Q}_\beta^{s,t}$ with $\mathbf{s} = \mathbf{0}$, the corresponding point in the diffeomorphic manifold is $\psi(\mathbf{x}) = \begin{pmatrix} \sqrt{|\beta|} \frac{\mathbf{t}}{\|\mathbf{t}\|} \\ \mathbf{0} \end{pmatrix}$. Based on the definition of tangent space, for any tangent vector $\boldsymbol{\xi} \in \mathcal{T}_\mathbf{x}\mathcal{Q}_\beta^{s,t}$, $\langle \mathbf{x}, \boldsymbol{\xi} \rangle_t = 0$ implies $-\mathbf{t}\boldsymbol{\xi}_t + \mathbf{0}\boldsymbol{\xi}_s = 0$, which means $\mathbf{t}\boldsymbol{\xi}_t = 0$. Thus, $\langle \psi(\mathbf{x}), \boldsymbol{\xi} \rangle_t = 0$. Based on the definition of tangent space, $\boldsymbol{\xi} \in \mathcal{T}_{\psi(\mathbf{x})}(\mathbb{S}_{-\beta}^t \times \mathbb{R}^s)$. $\square$

### D.4 Proof of Theorem 4

**Theorem 4.** *For any point* $\mathbf{x} \in \mathcal{Q}_\beta^{s,t}$*, the union of the normal neighborhood of* $\mathbf{x}$ *and the normal neighborhood of its antipodal point* $-\mathbf{x}$ *cover the entire manifold. Namely,* $\mathcal{U}_\mathbf{x} \cup \mathcal{U}_{-\mathbf{x}} = \mathcal{Q}_\beta^{s,t}$.

*Proof.* For any point $\mathbf{x} \in \mathcal{Q}_\beta^{s,t}$ and $\mathbf{y} \notin \mathcal{U}_x$. Based on the definition of normal neighborhood, $\langle \mathbf{x}, \mathbf{y} \rangle_t \geq |\beta| \rightarrow \langle -\mathbf{x}, \mathbf{y} \rangle_t \leq |\beta|$. Thus, $\mathbf{y} \in \mathcal{U}_{-x}$, and $\mathcal{U}_\mathbf{x} \cup \mathcal{U}_{-\mathbf{x}} = \mathcal{Q}_\beta^{s,t}$. $\square$

## E  Supplementary experimental details

In this section, we present the details of our experiments and implementations, as well as some additional analyses.

### E.1  Graph Reconstruction

For graph reconstruction, one straightforward method is to minimize the graph distortion [33, 20] given by

$$\frac{1}{|V|^2} \sum_{u,v} \left( \left( \frac{d(\mathbf{u}, \mathbf{v})}{d_G(u,v)} \right)^2 - 1 \right)^2, \tag{17}$$

where $d(\mathbf{u}, \mathbf{v})$ is the distance function in the embedding space and $d_G(u, v)$ is the graph distance (the length of shortest path) between node $u$ and $v$, $|V|$ is the number of nodes in graph. The objective function is to preserve all pairwise graph distances. Motivated by the fact that most of graphs are partially observable, we minimize an alternative loss function [24, 40] that preserves local graph distance, given by,

$$\mathcal{L}(\Theta) = \sum_{(u,v) \in \mathcal{D}} \log \frac{e^{-d(\mathbf{u}, \mathbf{v})}}{\sum_{\mathbf{v}' \in \mathcal{E}(u)} \exp^{-d(\mathbf{u}, \mathbf{v}')}}, \tag{18}$$

where $\mathcal{D}$ is the connected relations in the graph, $\mathcal{E}(u) = \{v | (u, v) \notin D \cup u\}$ is the set of negative examples for node $u$, $d(\mathbf{u}, \mathbf{v})$ is the distance function in the embedding space. For evaluation, we apply the mean average precision (mAP) to evaluate the graph reconstruction task. mAP is a local metric that measures the average proportion of the nearest points of a node which are actually its neighbors in the original graph. The mAP is defined as Eq. (19).

$$\mathrm{mAP}(f) = \frac{1}{|V|} \sum_{u \in V} \frac{1}{\deg(u)} \sum_{i=1}^{|\mathcal{N}_u|} \frac{|\mathcal{N}_u \cap R_{u,v_i}|}{|R_{u,v_i}|}, \tag{19}$$

where $f$ is the embedding function, $|V|$ is the number of nodes in graph, $\deg(u)$ is the degree of node $u$, $\mathcal{N}_u$ is the one-hop neighborhoods in the graph, $\mathbb{R}_{u,v}$ denotes whether two nodes $u$ and $v$ are connected.

**Implementation details.** We implement our models by Pytorch and *Geoopt* tool[3]. Our models and baselines are built upon the codes[4]. For baselines of product space and $\kappa$-GCN, we use the code[5]. Our models are trained on NVIDIA A100 with 4 GPU cards. Similar to [32], we use the pre-tranined embeddings on link prediction as the initialization of node classification, to preserve graph structures information. All non-Euclidean methods do not use attention mechanism for a fair comparison. For node classification and link prediction, the optimal hyper-parameters for all methods are obtained by the same grid search strategy. The ranges of grid search are summarized in Table 6.

Table 6: The grid search space for the hyperparameters.

| Hyperparameter | Search space |
| --- | --- |
| Number of layers | 1,2,3 |
| Weight decay | 0, 1e-3, 5e-4,1e-4 |
| Dropout rate | 0,0.1,0.2,0.3,0.4,0.5,0.6,0.7 |
| Activation | relu, tanh, sigmoid, elu |

**Numerical instability.** In practice, we find that the numerical computation in pseudo-hyperboloid is not stable due to the limited machine precision and rounding. To ensure the learned embeddings and tangent vectors to remain on the manifold and tangent spaces, we conduct projection operations via Eq. (20) and Eq. (21) after each vector operations.

$$\varphi(x) = \psi^{-1} \circ \psi, \tag{20}$$

where $\psi$ can be instantiated by Theorem 2. Note that $\psi(\cdot)$ is an identity function when $\mathbf{x}$ is already localted in $\mathcal{Q}_{\beta}^{s,t}$. For input $\mathbf{x} \notin \mathcal{Q}_{\beta}^{s,t}$, applying $\psi(\cdot)$ can project the $\mathbf{x}$ back to $\mathcal{Q}_{\beta}^{s,t}$, from which $\psi(\cdot)$ is no longer an identity.

$$\Pi_{\mathbf{x}}(\mathbf{z}) = \mathbf{z} - \frac{\langle \mathbf{z}, \mathbf{x} \rangle_t}{\langle \mathbf{x}, \mathbf{x} \rangle_t} \mathbf{x}, \tag{21}$$

where $\mathbf{z}$ is the tangent vector to be projected.

**Trainable curvature.** Similar to the trainable curvature in the hyperbolic space [32], we give an analogue of Theorem 4.1 in [32] as following corollary.

**Corollary 1.** *For any two points* $\mathbf{x}, \mathbf{y} \in \mathcal{Q}_{\beta}^{s,t}$, *there exists a mapping* $\phi : \mathcal{Q}_{\beta}^{s,t} \to \mathcal{Q}_{\beta'}^{s,t}$ *that maps* $\mathbf{x}, \mathbf{y}$ *into* $\mathbf{x}', \mathbf{y}' \in \mathcal{Q}_{\beta'}^{s,t}$, *such that the pseudo-Riemannian inner product is scaled by a constant factor* $\frac{\beta'}{\beta}$. *Namely* $\langle \mathbf{x}, \mathbf{y} \rangle_t = \frac{\beta'}{\beta} \langle \mathbf{x}', \mathbf{y}' \rangle_t$. *The mapping is given by* $\phi(\mathbf{x}) = \sqrt{\frac{\beta'}{\beta}} \mathbf{x}$.

---

[3]https://github.com/geoopt/geoopt
[4]https://github.com/HazyResearch/hgcn, https://github.com/CheriseZhu/GIL
[5]https://github.com/fal025/producthgcn

Corollary 1 shows that assuming infinite machine precision, pseudo-hyperboloids of varying curvatures should have the same expressive power. However, due to the limited machine precision [32], we set the curvature $1/\beta$ as a trainable parameter to capture the best scale of the embeddings.

**Skip connection.** In preliminary experiments of graph reconstruction, we found that the performance of GCN, HGCN, $\kappa$-GCN and $\mathcal{Q}$-GCN could not compete with the method in [20] that directly optimizes the distance function. We hypothesized that this was probably caused by over-smoothing of graph convolution kernel. Therefore, we decided to apply skip-connection strategy [49, 50, 51] to avoid oversmoothing. Specifically, we take the mean of the hidden output of each layer to the decoder. In the paper, we report the graph reconstruction results of GCN, HGCN, $\kappa$-GCN and $\mathcal{Q}$-GCN applied with skip-connection.

**Dataset analysis.** Table 7 and Table 8 summarize the statistics of the datasets we used for our experiments. For each dataset, we report the number of nodes, the number of edges, the number of classes and the feature dimensions. Besides, we measure the graph sectional curvature to identify the dominating geometry of each dataset. Fig. 4 shows the histograms of sectional curvature and the mean sectional curvature for all datasets. It can be seen that all datasets have both positive and negative sectional curvatures, showcasing that all graphs contain mixed geometric structures. However, the datasets possess a preference for the negative half sectional curvature except the case of Facebook that has dominating positive sectional curvature, showcasing that most of real-world graphs exhibit more hierarchical structures than cyclic structures.

To further analyze the degree of hierarchy of each graph, we apply $\delta$-hyperbolicity to identify the tree-likeness. Table 9 shows the $\delta$-hyperbolicity of different datasets. It shows that although most of the datasets have negative mean sectional curvature, their degrees of hierarchy vary significantly. For example, Airport and Pubmed are more hierarchical than CiteSeer and Cora.

Table 7: A summary of characteristics of the datasets for graph reconstruction.

| Dataset | Power | Bio-Worm | Web-Edu | Facebook |
|---|---|---|---|---|
| #Nodes | 4941 | 2274 | 3031 | 4039 |
| #Edges | 6594 | 78328 | 6474 | 88234 |

Table 8: A summary of characteristics of the datasets for node classification and link prediction.

| Dataset | Airport | Pubmed | CiteSeer | Cora |
|---|---|---|---|---|
| #Nodes | 3188 | 19717 | 3327 | 2708 |
| #Edges | 18631 | 44338 | 4732 | 5429 |
| #Classes | 4 | 3 | 6 | 7 |
| #Features | 4 | 500 | 3703 | 1433 |

Table 9: The $\delta$-hyperbolicity distribution of all used datasets.

| Datasets | 0 | 0.5 | 1.0 | 1.5 | 2.0 | 2.5 |
|---|---|---|---|---|---|---|
| Power | 0.4025 | 0.1722 | 0.1436 | 0.0773 | 0.0639 | 0.0439 |
| Bio-Worm | 0.5635 | 0.3949 | 0.0410 | 0 | 0 | 0 |
| Web-Edu | 0.9532 | 0.0468 | 0 | 0 | 0 | 0 |
| Facebook | 0.8209 | 0.1569 | 0.0221 | 0 | 0 | 0 |
| Airport | 0.6376 | 0.3563 | 0.0061 | 0 | 0 | 0 |
| Pubmed | 0.4239 | 0.4549 | 0.1094 | 0.0112 | 0.0006 | 0 |
| CiteSeer | 0.3659 | 0.3538 | 0.1699 | 0.0678 | 0.0288 | 0 |
| Cora | 0.4474 | 0.4073 | 0.1248 | 0.0189 | 0.0016 | 0.0102 |

Algorithm 1 describes the training process of the proposed method.

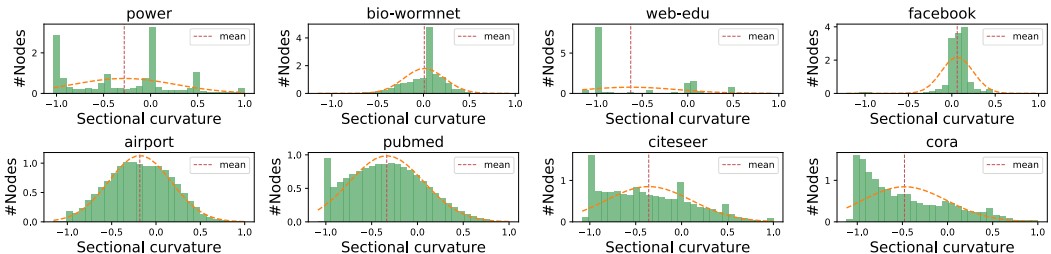

Figure 4: The histograms of sectional curvature for all used datasets.

---

**Algorithm 1** The pseudo-code of $\mathcal{Q}$-GCN

---

**Input:**

    The input graph $\mathcal{G} = (\mathcal{V}, \mathcal{E})$, the input node features $\mathbf{x}$, the time dimension $t$, the space dimension $s$, the initial curvature $\beta$, the number of epoches $E$, the number of layers $L$, the initial learning rate, the activation function $\sigma$, the layer-wise trainable weight $\mathbf{W}$ and bias term $\mathbf{b}$.

**Output:**

    The trained model, i.e., the parameters $\mathbf{W}$, $\mathbf{b}$ and $\beta$. The resulting embeddings $\mathbf{h}$ of nodes.

  1: **Feature initialization:** map the input Euclidean features $\mathbf{x}$ into Pseudo-Riemannian space by the double projection $\mathbf{h}^0 = (\psi^{-1} \circ \psi)(\mathbf{x})$ where $\psi$ and $\psi^{-1}$ are given by $\quad\quad\quad\triangleright Eq.\ (5)$.

  2: **for** $e \in [0, E)$ **do**

  3:     **for** $l \in [0, L)$ **do**

  4:         **for** $i \in \mathcal{V}$ **do**

  5:           $\mathbf{h}_i^\ell = W^\ell \otimes^\beta \mathbf{h}_i^\ell := \widehat{\exp}_{\mathbf{o}}^\beta \left( W^\ell \widehat{\log}_{\mathbf{o}}^\beta \left( \mathbf{h}_i^\ell \right) \right)$ (**Feature transformation**)

  6:           $\mathbf{h}_i^\ell = \widetilde{\mathbf{h}}_i^\ell \oplus^\beta \mathbf{b}^\ell := \begin{cases} \exp_{\widetilde{\mathbf{h}}_i^\ell}^\beta \left( P_{\mathbf{o} \to \widetilde{\mathbf{h}}_i^\ell}^\beta \left( \mathbf{b}^\ell \right) \right), & \text{if } \left\langle \mathbf{o}, \widetilde{\mathbf{h}}_i^\ell \right\rangle_t < |\beta| \\ -\exp_{-\widetilde{\mathbf{h}}_i^\ell}^\beta \left( P_{\mathbf{o} \to -\widetilde{\mathbf{h}}_i^\ell}^\beta \left( \mathbf{b}^\ell \right) \right), & \text{if } \left\langle \mathbf{o}, \widetilde{\mathbf{h}}_i^\ell \right\rangle_t \geq |\beta| \end{cases}$ (**Bias**

           **translation**)

  7:           $\mathbf{h}_i^{\ell+1} = \widehat{\exp}_{\mathbf{o}}^{\beta_{\ell+1}} \left( \sigma \left( \sum_{j \in \mathcal{N}(i) \cup \{i\}} \widehat{\log}_{\mathbf{o}}^{\beta_\ell} \left( \mathbf{h}_j^\ell \right) \right) \right)$ (**Neighborhood aggregation**)

  8:         **end for**

  9:         //Computing the task-dependent loss $\mathcal{L}$.

10:         //Updating the model parameters $\mathbf{W}$, $\mathbf{b}$, and $\beta$ using Adam optimizer.

11:         **if** convergence **then**

12:           break loop.

13:         **end if**

14:     **end for**

15: **end for**

16: //Task-dependent inference.

17: **return** $\mathbf{W}$, $\mathbf{b}$, $\beta$ and $\mathbf{h}$.

---