# OpenReview forum: "Pseudo-Riemannian Graph Convolutional Networks"
_NeurIPS.cc/2022/Conference — NeurIPS 2022 Accept_

### Official Review · Reviewer_qApk · 2022-07-06

**Rating:** 7
**Confidence:** 4
**Soundness:** 3 good
**Presentation:** 3 good
**Contribution:** 3 good

**Summary:**

The authors extend existing Graph Convolutional Networks (GCNs) on non-Euclidean Riemannian manifolds to a larger class of pseudo-Riemannian manifolds which generalizes hyperboloid and sphere. They extend neural network operations into geodesically disconnected pseudo-Riemann manifolds by developing some new geodesic tools, and derive a pseudo-Riemannian GCN (Q-GCN) which models data in pseudo-Riemannian manifolds of constant nonzero curvature. This method provides a geometric inductive bias that captures heterogeneous topologies. The representational capabilities are examined through experiments on graph reconstruction, node classification and link prediction tasks. The empirical results show that the proposed method outperforms Euclidean GCNs and non-Euclidean GCN counterparts with an acceptable increase in time complexity.

**Questions:**

Please refer to the weakness above.

**Limitations:**

The authors have adequately addressed the limitations of the proposed method, such as higher computational costs, and the results on graphs with dominating hierarchical properties.

**Strengths And Weaknesses:**

Strengths:
- The work is original: there has not been any existing works about GCNs on pseudo-Riemannian manifolds.
- The presentation of the paper is clear.
- The paper includes adequate theoretical derivations on how the geodesic tools are developed, and empirical results to support that the proposed Q-GCN accommodates topologically heterogeneous graphs well, and outperforms other existing methods, either with Euclidean GCNs or non-Euclidean GCNs, on most of the datasets.

Weaknesses:
- It would be better to include some visualizations either for illustrating the theoretical part or presenting the experimental results.

---

> ### Author Response · Authors · 2022-08-02
> **Response to reviewer qApk**
>
>
> Thank you so much for your positive review. As you suggested, we added a visualization section for the learned graph embeddings in the updated Appendix.

---

> > ### Comment · Reviewer_qApk · 2022-08-08
> > **Thanks for the response**
> >
> > I don't have further questions and will keep the current rating.

---

### Official Review · Reviewer_Pmuy · 2022-07-09

**Rating:** 7
**Confidence:** 4
**Soundness:** 4 excellent
**Presentation:** 3 good
**Contribution:** 4 excellent

**Summary:**

This paper proposes a non-Euclidean GCN based on pseudo-Riemannian geometry. The major idea is to generalise GCN operations including feature transformation and neighbourhood aggregation into pseudo-Riemannian manifolds. Different from Riemannian manifolds such as sphere and hyperboloid, the main challenge is the broken geodesic issue (i.e., there exists broken geodesic for which standard operations are not well defined). The authors propose to use “diffeomorphism operations” to define basic operations like linear transformation and aggregation. Experiment results on three graph learning benchmarks show significant improvements.

**Questions:**

1.Is it possible to directly perform aggregation in the pseudo-hyperboloid, like using  Frechet mean to simplify the aggregations as done by Y. Zhang, et al. 2021.?
2.The authors use tangent space optimization. Is there any guarantee for doing so?

[1] Y. Zhang, et al. Lorentzian Graph Convolutional Networks. 2021.

**Limitations:**

Yes, the authors have highlighted the limitations of their work in the Broder Impact section

**Strengths And Weaknesses:**

Strengths
1.The authors extended GCN into pseudo-Riemannian manifolds (more specifically, pseudo-hyperboloid) with non-positive definite metric. The motivation is inspired by Marc T. Law, et al. 2020. However, different from Marc T. Law, et al. 2020, the authors propose to generalize the neural network operations into the pseudo-hyperbolic space.
2. The proposed operations and tools seem technical solid. The main non-trivial part is that there exists broken geodesic for which standard manifold operations like expmap and logmap cannot be defined.  The authors propose to map the operations into diffeomorphic manifolds that are easy to operate. The proposed tools might be useful for generalizing other neural models to pseudo-hyperboloid like generative models.
3.Evaluation results on three graph learning tasks are strong enough to show the benefits of the proposed method on embedding graphs, especially when the graphs have mixed curvature.
Weaknesses
There are a few minor issues that could be fixed during the rebuttal
1. The authors should give enough details of the baselines and explain why choosing them
2. For a given dataset, how to decide the proper settings of time dimension and why choosing those time dimensions (2,4,8,…) in the evaluation is not clear enough

---

> ### Author Response · Authors · 2022-08-02
> **Response to reviewer Pmuy**
>
> **Details/choices of the baselines.** The baselines are chosen to reflect the focus of work, i.e., to validate whether the proposed pseudo-Riemannian GCN outperforms previous Euclidean (GCN, GAT, SAGE, etc.) and non-Euclidean baselines (HGCN and $\kappa$-GCN). To the best of our knowledge, HGCN and $\kappa$-GCN are the most SoTA non-Euclidean counterparts. We will add more details about the compared baselines.
>
> **The choice of time dimension.**  Essentially, for a given dataset, we could use some graph geometric metric such as $\delta$-hyperbolicity and graph curvature to identify the tree-likenes (see Fig.~3 and Table 9 for a full analysis of dataset). We then could adjust the time dimension based on the graph geometric metrics, e.g., the lower the $\delta$-hyperbolicity is, the fewer time dimensions we use. We choose time dimensions (1,3,5,…) to study how the choice of time dimensions influences the performance.
>
> **Fréchet mean in pseudo-hyperboloid.** We agree that Fréchet mean should be a better approximation of the mean in manifolds in general. However, unlike hyperbolic manifold that has a closed-form of Fréchet mean, the calculation of Fréchet mean in pseudo-hyperboloid requires iterative optimization, which is much more computationally expensive than tangent mean.
>
> **Tangent space optimization.** Note that the tangent space optimization has theoretical guarantee, if and only if there is a global bijection between the manifold to the tangent space. Which is exactly our case due to the diffeomorphic operations we designed.

---

### Official Review · Reviewer_zhRd · 2022-07-11

**Rating:** 4
**Confidence:** 1
**Soundness:** 2 fair
**Presentation:** 3 good
**Contribution:** 2 fair

**Summary:**

The authors extended the graph convolutional networks to the graph representation into pseudo Riemannian geometry. Pseudo-Riemannian geometry was introduced as a possible embedding space for hierarchical graphs and graphs with cycles. As the space is equipped with metric, geodesic, exp and log map, the authors extend the graph convolutional neural network model to such framework. The model performance is showcased on both graph reconstruction data, link prediction and node classification.

**Questions:**

Introduction: For a reader without a strong knowledge on graph embedding, it is not very clear the motivation of the work. Why such complex embedding is required for graph embedding? Authors should help the reader understanding the literature gap they are trying to fill, maybe adding a simple example in a sentence.
14 what does it means graphs with different curvature? The curvature of the embedding space? In geometry, the curvature is a property of the metric. This sentence is not so clear as the reader might be more familiar with the metric curvature rather than the graph curvature. Notice that this is an important point to be clarified throughout the paper.
29-30 I don’t understand why the authors cite these examples. Is the spherical embedding another embedding for graphs? If so should be better explained as it is the only other example of non-Euclidean data type for which Neural Networks have been extended.

Section 3 The theory described in Section 3 strongly relies on [21]. I think the novelty should be better enlighten and justified. Is the Theorem 2 needed to extend the GNN? Why?

Subsection 3.2: Tangential Transformation. I don’t understand a tangential approximation in a point should be a good approximation of the data. Especially in the example with positive curved spaces, the tangent space centered in a point could be a bad approximation of your data if there is no ensurance of data concentration. A better justification should be provided.

Experiments: How do the authors justify a 100% mean Average Precision in Table 1?


**Limitations:**

The paper is complex and extend the GNN to a pseudo-Riemannian embedding of Graphs. As the geometrical embedding is fully described by [21], the paper should be better framed to justify which geometrical tools are necessary to extend GNN and why this complex framework is needed compared to more intuitive GNN on the Euclidean embedding or Hyperbolic embedding of graphs.

**Strengths And Weaknesses:**

The authors focus on an important topic which is the extension of the neural network models to a geometrical framework beyond the Euclidean space or any other Smooth Manifold. Geometrical setting is well introduced and the geometrical tools are well described. They showcase the model in three important tasks related to graph analysis.
The main strengths are: (1) the attention to a careful geometrical embedding of the data (2) a well-structured paper (3) complete experiments.
The main weakness is lack of novelty compare to [21] and the complexity of the framework, which it is not well justified throughout the paper.

The authors are extending the GNN to the pseudo-Riemannian setting, but the geometrical tools seem to be the one already introduced in [21].  If the main difference is the spherical mapping described in Th. 2, I don't understand why a map preserving the curvature should be preferable to a map with the same properties but with a flat curvature image (which is easier to handle).
In Section 4, the authors describe how to extend the GNN to the framework by mapping node features (in the Euclidean setting) to the described embedding and then mapping them again into the tangent space (which is flat by definition). Is it a geometrical virtuosism or a necessity? In addition, there is a tangent space approximation centered in the south pole described in section 3.2. The tangent space approximation is a bad approximation of positively curved spaces if there is no ensurance about how the data are concentrated. This step is vague and unclear to me.
Lastly, there should be a stronger justification to the introduction of such complex (and parametric -see time dependence) setting. As there are many different embedding graphs procedure, which are the strenghts of such embedding? This should be better enlightened in both the introduction and the experiments, as not all the reader might be familiar with such embedding.

[21] Law and Stam (2020) Ultra hyperbolic Representation Learning.

---

> ### Author Response · Authors · 2022-08-02
> **Response to reviewer zhRd (part 1)**
>
> **Motivation of our work (justification of such complex embeddings, why cite spherical embeddings).**  Our work is motivated by the fact that different graphs are suitable to be embedded into different manifolds.
> For instance, graphs with hierarchies (e.g., citation networks) can be well encoded into a (negatively curved) hyperbolic space, while graphs with cycles (e.g., transportation networks) can be well encoded into a (positively curved) spherical space.
> Our work unifies these seemingly contradictory requirements by encoding graphs into a pseudo-Riemannian manifold that generalizes hyperbolic and spherical manifolds, so that it can simultaneously deal with hierarchies and cycles for modeling more complex graphs (i.e., hierarchies with cycles. Actually, most real-world graphs contain both hierarchies and cycles). This is also why we discuss both hyperbolic and spherical embeddings in our introduction.
>
> **What is meant by graphs with different curvature?**
> We are talking about discrete curvature (e.g., graph sectional curvature) which is studied in many previous works such as in Ze Ye et al. A graph with positive (resp. negative) curvature means the graph is more spherical (resp. hierarchical). We will clarify them in our revision by saying "graphs with mixed topologies" to avoid misunderstanding. Thank you so much for this suggestion.
>
> [1] Ze Ye, et al. Curvature graph network. ICLR 2019.
>
> **Necessity of feature initialization.**
> The initial node features are defined in a normal Euclidean space but not in any specific tangent space of the manifold. Hence, our feature initialization in sec.3.2 is a \textbf{necessary} step. Of course, one could initialize node features in two ways: 1) mapping to the manifold or 2) mapping to a tangent space of the manifold. We applied 1) because our GCN updating formula in each layer in Eq(6) assumes that the features are defined in the manifold (otherwise if we use method 2) we have to distinguish the formula of the first layer with the formulas of other layers). See our pseudo-code in the Appendix.
>
> **Necessity of theory 2 for extending GNN.**
> Our theorem 2 is a necessary condition for our theorem 3, while theorem 3 is a necessary condition for generalizing GNN operations into pseudo-Riemannian manifolds (pseudo-hyperboloid). The reason is that the diffeomorphism in Theorem 2 preserves the curvatures of the diffeomorphic components (e.g., the spherical component has curvature $|\beta|$).
> It is necessary that the curvature is preserved such that mapped points still lie on the surface of the pseudo-hyperboloid and the diffeomorphic manifold share the same tangent space when the space dimension=$\mathbf{0}$ (see our theorem 3). Hence, although we use the diffeomorphism, our final operations are actually performed in the tangent space of the original pseudo-hyperboloid manifold (not only in the product manifold). However, if we just use theorem 1 in [21], we will no longer have such properties and we have to do all operations in the tangent space of a product manifold that is not identical to the tangent space of the original manifold.
>
> **Theoretical contribution.**  Our theoretical contributions are clarified in line 198-204: besides that we develop theorems 2 \& 3 (which are used as necessary condition for formulating tangential operations of pseudo-hyperboloid to avoid broken cases), we re-formulate parallel transport with Theorem 4 to avoid issues with broken geodesics
> and propose a new distance measure using the broken geodesic in Eq.(4), which is different from the approximated distance in [21]. Both are also useful components for generalizing GNNs to pseudo-Riemannian manifolds.

---

> ### Author Response · Authors · 2022-08-02
> **Response to reviewer zhRd (part 2)**
>
> **Justification of tangential transformation (especially in positive curved spaces).** Note that for positively curved (spherical) manifold, we do not have any assumption on the data concentration. This is because spherical manifold is a symmetric space and it does not matter where the data is centered on the sphere. For example, data centered on the north pole (or any other points) can be isometrically moved to be centered on the south pole without losing any distance information (we only care about the embedding distance info but not where the embeddings are centered). In general, tangential transformation is a good approximation if and only if there is a global bijection from the tangent space to the manifold, which is exactly our case as well as the cases of hyperbolic and spherical manifolds.
>
> **100\% mean Average Precision in Table 1.** Actually, mAP (which are also used in [2,3] for graph reconstruction) is a local metric that measures the average proportion of the nearest points of a node which are actually its neighbors in the original graph (see Eq.(19) in our Appendix E.1 for details). In other words, mAP does not track explicit pair-wise distances, but rather a ranking-based measure for local neighborhoods. $100\%$ mAP for graph reconstruction basically means that our resulting embeddings preserve all local neighborhood information (but not necessary all pair-wise distance information). Note also that in graph embeddings, preserving neighborhood information is more important than preserving all pair-wise distance information, which is why we (and also [2,3]) choose mAP as our measures. The mAP for graph reconstruction on Power datasets is $1.000$ with $3$ decimal places (averaged by 10 random seeds).
>
> [2] Maximillian Nickel and Douwe Kiela.  Poincaré embeddings for learning hierarchical representations. NeurIPS 2017.
>
> [3] Albert Gu, et al.  Learning mixed-curvature repre-428sentations in product spaces.  ICLR 2019.

---

### Official Review · Reviewer_vzK2 · 2022-07-11

**Rating:** 7
**Confidence:** 3
**Soundness:** 3 good
**Presentation:** 3 good
**Contribution:** 3 good

**Summary:**

To unify the successes of hyperbolic graph convolutional networks (GCN) in modeling hierarchical structures and spherical manifolds in modeling cycles,
the authors generate a diffeomorphism $\psi$ from pseudo-hyperboloids ( $Q_\beta^{(s,t)}$ with $t+1$ time dimensions and $s$ spatial dimensions) into product manifolds of a sphere and Euclidean space. The special form of this diffeomorphism allows them to formulate expMap, logMap, parallel transport operations and geodesics to avoid broken cases (using antipodal points). The resulting Q-GCNs have learnable layer-wise parameters in Euclidean space and can be used for both graph reconstruction (GR) and link prediction (LP) / node classification (NC) tasks. Q-GCNs outperform competitors such as $\kappa$-GCN's and HGCN's on these tasks on a majority of the datasets (Tables 1 and 2). The authors also conduct a thorough parameter sensitivity analysis and computation time analysis (Tables 5 and 4).


**Questions:**

The authors mention at the end of the Optimization section that dedicated optimization like [21] would be considered in future work.
What advantage would dedicated optimization provide ?

Please note the questions inherent in Weaknesses 1 and 2 above.


**Limitations:**


The authors mention that improvements might be necessary for automatically choosing time dimensions of Q-GCNs in Sec. 5


**Strengths And Weaknesses:**

Strengths
1. As summarized above, the approach is motivated quite well both from an intuitive and from a theoretical viewpoint.
2. Q-GCNs outperform competitors such as $\kappa$-GCN's and HGCN's on GR or LP/NC tasks on a majority of the datasets (Tables 1 and 2)
3. The authors also conduct a thorough parameter sensitivity analysis and computation time analysis (Tables 5 and 4).
4. Q-NNs, another variant introduced using the authors' theoretical analysis, outperforms MLPs on NC tasks in Table 3.

Weaknesses

1. Some of the theorems reported by the authors are quite technical and require significant background on pseudo-Riemannian manifolds. But it is not clear how novel these results would be considered in those fields. For instance, is the proof for theorem 2 is significantly novel w.r.t. the proof of theorem 1 in [21] ? It is also not clear why the discontinuous operation in (4) doesn't sacrifice the diffeomorphic property. It's possible parallel transport doesn't require the continuity property, but this isn't explicitly pointed out.

2. The authors do not provide pseudo-code for their training and inference algorithms either in the main body of the paper or in the Appendix.
So it is not entirely obvious to a reader when exactly the antipodal operation described after Theorem 4 or in the bias translation subsection of Sec. 3.2 is applied, although this can be inferred from (6).

3. Error bars are reported in tables 2 and 3, but not in table 1.

---

> ### Author Response · Authors · 2022-08-02
> **Response to reviewer vzK2**
>
> **Necessity and novelty of theorem 2.** Our theorem 2 is a necessary condition for our theorem 3, while theorem 3 is a necessary condition for generalizing GNN operations into pseudo-Riemannian manifolds (pseudo-hyperboloid). The reason is that the diffeomorphism in Theorem 2 preserves the curvatures of the diffeomorphic components (e.g., the spherical component has curvature $|\beta|$).
> It is necessary that the curvature is preserved such that mapped points still lie on the surface of the pseudo-hyperboloid and the diffeomorphic manifold share the same tangent space when the space dimension=$\mathbf{0}$ (see our theorem 3). Hence, although we use the diffeomorphism, our final operations are actually performed in the tangent space of the original pseudo-hyperboloid manifold (not only in the product manifold).
> However, if we just use theorem 1 in [21], we will no longer have such properties and we have to do all operations in the tangent space of a product manifold that is not identical to the tangent space of the original manifold.
>
> The proof of theorem 2 is different from the proof of theorem 1 in [21]. Our proof is inspired by a geometric intuition that each point in the pseudo-hyperboloid can be projected onto a spherical conic section (a spherical submanifold centered around the origin) of the pseudo-hyperboloid whose space dimenion is $\mathbf{0}$. The points in the spherical conic section correspond to the special reference points that satisfy theorem 3.
>
> **Theoretical contribution.** Our theoretical contributions are clarified in line 198-204: besides that we develop theorems 2 \& 3 (which are used as necessary condition for formulating tangential operations of pseudo-hyperboloid to avoid broken cases), we re-formulate parallel transport with Theorem 4 to avoid issues with broken geodesics
> and propose a new distance measure using the broken geodesic in Eq.(4), which is different from the approximated distance in [21]. Both are also useful components for generalizing GNNs to pseudo-Riemannian manifolds.
>
>
>
> **Why discontinuous operation in eq(4) doesn't sacrifice the diffeomorphic property?** Actually, our piecewise distance function in eq(4) is a continuous function. In the special case where $<x,y>_t = |\beta|$ (which corresponds to the critical point of the g-disconnected cases), the distance corresponds to a half of the perimeter of a sphere, i.e., $\pi \sqrt{|\beta|}$. For g-disconnected cases, our distance function is defined by an approximation of broken geodesics (e.g., two piece-wise geodesics).
> Note that in differential geometry, when a geodesic does not exist, using broken geodesics as approximation is a standard method.
>
> **Pseudo-code for the training and inference.** We add pseudo-code for QGCN in the Appendix. Thank you so much for the suggestion.
>
> **Error bars in table 1:** As we explained in line 299, all results are averaged over $10$ runs with random seeds. However, the standard deviations on graph reconstruction (Table 1) are relatively small and are hence omitted for simplicity. We will point out this explicitly in our revision.
>
> **What advantage would dedicated optimization provide?** Although tangent space optimization works well, the resulting gradient direction of SGD can be viewed as an approximation of the gradient direction in the manifold. We believe that dedicated optimization might further improve the convergence speed but this requires empirical validation.

---

> > ### Comment · Reviewer_vzK2 · 2022-08-09
> > **Satisfactory response**
> >
> > The authors have adequately addressed my concerns and questions.
> > I wish to thank them for the refresher / clarification on the use of broken geodesics for approximation and for the insightful comment regarding Frechet means to another reviewer.
> > My rating remains unchanged after considering the discussion between authors and reviewers thus far.

---

### Author Response · Authors · 2022-08-08
**Any further questions**

We thank all reviewers again for spending valuable time reading our paper and for providing helpful comments and suggestions. We would like to ask whether the reviewers have any further comments or suggestions, and we are happy to discuss further with you. Thank you so much.

---

### Meta-Review · Area_Chair_GPJV · 2022-08-26

**Recommendation:** Accept
**Confidence:** Certain

**Metareview:**

This paper is about using certain types of non-Euclidean spaces for representations. The background context for the paper is roughly the following: hyperbolic spaces have been recently popularized for embedding tree-like data. These are one of the three types of so-called constant curvature Riemannian manifolds. More recently, to handle more types of data, products of manifolds were proposed. This is more flexible, as such spaces can represent many more flavors of data. The authors of the present work further generalize these embedding spaces by using pseudo-Riemannian manifolds, which, unlike Riemannian manifolds, do not require positive semidefinite metrics.

The hope for these types of spaces is that they are even more flexible, capable of representing potentially any type of structure. The cost is that they have weird behavior, and that the operations developed for more specific spaces do not easily lift. The authors derive these various operations and build equivalents of GNNs in these spaces, performing comparisons that are favorable against some of the existing literature.

This paper's strengths are that it does heavy technical work to get these fairly complicated spaces to work. The idea has potential, and there's plenty of additional interesting questions that result from examining these spaces and models built over them.

Most of the reviewers were in agreement about the paper's contributions. One reviewer disagreed and asked some reasonable questions, but most of these are answerable, and I encourage the authors to carefully revise some of their writing to produce additional intuition that provides these answers. Ultimately, I believe the paper clears the bar.

I will note a few downsides that I think the authors should address in their next revision:
- The authors should more carefully compare to products of Riemannian manifolds. The current motivation isn't sufficient (the authors say "the simple combination of spaces still does not accommodate topologically heterogeneous graphs very well"). Are there canonical graphs where pseudo-Riemannian manifolds provide low-distortion embeddings and no product manifold does? Or some other type of evidence?

- Similarly, in the experiments, the comparison against k-GCN, which is probably the closest competitor since it effectively generalizes hGCN, should be attempted with more combinations of spaces (i.e., instead of just H^5 x S^5, you could do H^2 x S^8, H^4 x S^6, and so on). This would make the comparison more fair, as currently the authors provide many more q-GCN implementations. More generally, not sure why there isn't a hyperparameter search over the signature rather than fixing it a priori.



**Award:**

No

---

### Decision · Program_Chairs · 2022-09-14

Accept